

# Free fermions with dephasing and boundary driving: Bethe Ansatz results

Vincenzo Alba

Dipartimento di Fisica dell' Università di Pisa and
INFN, Sezione di Pisa, I-56127 Pisa, Italy

vincenzo.alba@unipi.it

## Abstract

By employing the Lindblad equation, we derive the evolution of the two-point correlator for a free-fermion chain of length $L$ subject to bulk dephasing and boundary losses. We use the Bethe Ansatz to diagonalize the Liouvillian $\mathcal{L}^{(2)}$ governing the dynamics of the correlator. The majority of its eigenvalues are complex. Precisely, $L(L-1)/2$ complex eigenvalues do not depend on dephasing, apart from a trivial shift. The remaining complex levels are perturbatively related to the dephasing-independent ones for large $L$. The long-time dynamics is governed by a band of real eigenvalues, which contains an extensive number of levels. They give rise to diffusive scaling at intermediate times, when boundaries can be neglected. Moreover, they encode the breaking of diffusion at asymptotically long times. Interestingly, for large loss rate two boundary modes appear in the spectrum. The real eigenvalues correspond to string solutions of the Bethe equations, and can be treated effectively for large chains. This allows us to derive compact formulas for the dynamics of the fermionic density. We check our results against exact diagonalization, finding perfect agreement.

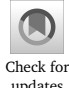
# 1 Introduction

Markovian master equations [1], such as the Lindblad equation, provide a versatile tool to understand the interplay between coherent and dissipative dynamics in *open* quantum many-body systems [2]. Although the interaction with an environment typically is a strong adversary for quantum coherence, it can also be exploited to imprint nontrivial quantum correlations [3], to aid quantum computation [4], or to stabilize topological order [5].

Exact solvable models could potentially help to build a general understanding of open quantum systems, similar to what happened in out-of-equilibrium *closed* systems [6]. Unfortunately, despite intense effort there are comparatively few examples of exact solvable Lindblad equations. Free-fermion and free-boson models subject to arbitrary *linear* jump operators lead to quadratic Liouvillians, and stand out as prominent examples [7]. Still, non quadratic Liouvillians that are solvable, for instance by the Bethe Ansatz, exist [8–24]. Remarkably, it has been shown in Ref. [11] that the Liouvillian describing the out-of-equilibrium dynamics of the fermionic tight-binding chain with global dephasing can be mapped to the Hubbard chain with imaginary interaction strength, which can be solved by Bethe Ansatz [25]. Interestingly, it is well-established that the dynamics of simple observables, such as few-point correlation functions can be obtained analytically [26–28], without explicitly relying on the exact solvability of the full Liouvillian. Furthermore, integrability is crucial to devise effective descriptions for out-of-equilibrium open systems. For instance, it has been shown recently that, by exploiting integrability, the Lindblad dynamics of paradigmatic observables can be captured within the hydrodynamic framework [29–34]. For quadratic Liouvillians one can employ the quasiparticle picture [35–37] to describe the dynamics of entanglement-related quantities [38–41]. Similar results were derived for free fermions in the presence of localized dissipation [42–44]. In conclusion, widening the set of integrable Lindblad equations is of paramount importance to make progress in out-of-equilibrium open quantum systems.

Here we provide new results in this direction considering the setup depicted in Fig. 1. We focus on a fermionic tight-binding chain of length $L$ with open boundary conditions. Global dephasing is present on each site of the chain. We denote by $\gamma$ the dephasing rate. Besides, the chain is subject to incoherent fermion losses with the same rate $\gamma^-$ at the edges.

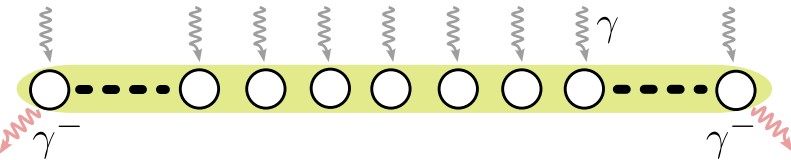

Figure 1: Setup considered in this paper. A tight-binding fermionic chain with $L$ sites is subject to global dephasing, with dephasing rate $\gamma$. We employ open boundary conditions. At the edges of the chain fermions are incoherently removed at rate $\gamma^-$.

This prototypical setup was investigated in Ref. [8], and more recently in Ref. [45, 46] (see also Ref. [47,48]) and [49,50]. The focus was on the interplay between the diffusive transport induced by the global dephasing and the ballistic one due to the boundary driving. The same setup was employed to study the interplay between dissipation and criticality [2, 51]. Here we focus on the two-point fermionic correlation function $G_{x_1,x_2} := \text{Tr}(\rho(t)c^\dagger_{x_1} c_{x_2})$, with $c_{x_1}, c_{x_2}$ standard fermion operators, and $\rho(t)$ the full-system density matrix. The out-of-equilibrium dynamics of $G_{x_1,x_2}$ is governed by a Liouvillian super operator $\mathcal{L}^{(2)}$ (see section 2).

We show that the spectrum, i.e., eigenvalues and eigenvectors, of $\mathcal{L}^{(2)}$ can be constructed explicitly using the Bethe Ansatz. This happens despite the fact that the full Liouvillian, to the best of our knowledge, is not integrable. Indeed, as we show, the full Liouvillian is mapped to the one-dimensional Hubbard model with imaginary density-density interaction, imaginary boundary magnetic fields, and imaginary boundary pair production. The last term creates a pair of fermions with opposite spins at the boundaries, making the model non integrable. Since $\mathcal{L}^{(2)}$ is not hermitian, its eigenvalues (energies) are in general complex. The spectrum of $\mathcal{L}^{(2)}$, at least for moderate dephasing rate, splits into three different components. We show that there are $L(L-1)/2$ complex eigenvalues that are trivially related by a shift by $-\gamma$ to the eigenvalues obtained in the absence of dephasing. These correspond to eigenstates that are free-fermionic in nature. For $\gamma^- = 0$ these eigenvalues form a vertical band in the complex plane, their real part being $-\gamma$. For nonzero $\gamma^-$ the band is deformed. Nearby in energy, there are $\sim L(L-1)/2$ complex eigenvalues, which become the same as the dephasing-independent ones in the large $L$ limit.

Finally, a band containing $\sim L$ real eigenvalues is present. A similar band is present for periodic boundary conditions (see Ref. [26] and Ref. [28]), where it is responsible for diffusive dynamics at long times. For this reason we dub it diffusive band. The eigenvalues with the largest real parts, and the gap of $\mathcal{L}^{(2)}$, are in the diffusive band. The diffusive band correspond to so-called string solutions of the Bethe equations, which in the large $L$ limit can be treated within the framework of the string hypothesis [25]. For instance, this allows us to derive the Liouvillian gap analytically as $\Delta\mathcal{L}^{(2)} = -2\pi^2/(\gamma L^2) + \beta/L^3 + \mathcal{O}(L^{-4})$, where the constant $\beta$, which we determine, depends both on $\gamma$ and $\gamma^-$. The number of eigenvalues in the diffusive band depends on $\gamma, \gamma^-$. We show that in the large $L$ limit, i.e., in the regime of validity of the string hypothesis, there is a "critical" $\gamma_c = 4$ above which the band contains the largest number of eigenvalues. Upon lowering $\gamma$ the band gets progressively depleted. In the limit $L \to \infty$ the energy $\varepsilon$ in the diffusive band are such that $\varepsilon > \sqrt{\gamma^2 - 16} - \gamma$ for $\gamma > \gamma_c$. The number of levels depends on $\gamma^-$ as well. Precisely, for $\gamma > \gamma_c$ the band contains $L$ levels for $\gamma^- < \gamma_c^- = \exp(-\text{arccosh}(\gamma/4))$. At larger $\gamma^-$, two of the eigenvalues detach from the diffusive band, and are pushed to lower $\text{Re}(\varepsilon)$ upon increasing $\gamma^-$. The splitting between them is exponentially suppressed with $L$. These eigenvalues correspond to edge modes of $\mathcal{L}^{(2)}$, and are reminiscent of the boundary-related eigenstates of the Hubbard chain with boundary magnetic fields [25].

The Bethe Ansatz diagonalization of $\mathcal{L}^{(2)}$ allows, in principle, to obtain the full-time dynamics of $G_{x_1,x_2}(t)$. This is not straightforward because it requires to extract all the $L^2$ solutions of

the Bethe equations. Still, in the long-time limit the dynamics of the correlator is determined by the diffusive band, which can be treated by using the string hypothesis. Here we provide compact expressions for the dynamics of the density profile starting from a fermion localized at an arbitrary site of the chain. This is the main ingredient to obtain the dynamics from an arbitrary initial density profile. At long times, but short enough that the boundaries can be neglected, the density profile exhibits the same diffusive scaling as for periodic boundary conditions. At long time the diffusive regime breaks down due to the boundary losses.

The manuscript is organized as follows. In section 2 we introduce the tight-binding chain with dephasing and boundary losses, and the Lindblad equation. In section 3 we present the Bethe Ansatz treatment of the Liouvillian $\mathcal{L}^{(2)}$. Specifically, in section 3.1 we introduce the Ansatz for the eigenstates of $\mathcal{L}^{(2)}$. In section 3.2 we derive the Bethe equations. In section 3.3 we discuss the eigenvalues that correspond to dephasing-independent solutions of the Bethe equations. In section 3.4 we investigate the eigenvalues of $\mathcal{L}^{(2)}$ that are perturbatively connected to the dephasing-independent ones in the large $L$ limit. Finally in section 3.5 we discuss the diffusive band. In section 4 we focus on the dynamics of the two-point fermionic correlation function. In particular, in section 4.1 we discuss how to expand the initial correlator in the basis of the Bethe states. In section 4.2 we derive the normalization of the Bethe states. In section 4.3, by using the string hypothesis, we derive the long-time limit of the fermion density profile. We discuss numerical results in section 5. Precisely, in section 5.1 we focus on the solution of the Bethe equations. We provide the full set of solutions for chains with $L = 2$ and $L = 3$. In section 5.2 we discuss the solution of the Bethe-Gaudin-Takahashi equation for the diffusive band. In section 5.3 we overview the general structure of the eigenvalues of $\mathcal{L}^{(2)}$ presenting exact diagonalization (ED) data. In section 5.4 we compare the ED data against Bethe Ansatz results. In section 5.5 we focus on the finite-size scaling of the Liouvillian gap. In section 5.6 we benchmark the Bethe Ansatz results for $G_{x_1,x_2}$ with exact diagonalization. Section 5.7 provides numerical results for the dynamics of the density profile. In section 5.8 we focus on the diffusive scaling of the fermion density and its violation due to the boundary losses. We conclude in section 6. In Appendix A we show that the full Liouvillian of the system is mapped to a one-dimensional Hubbard model with imaginary interaction, imaginary boundary fields and imaginary pair-production term at the boundary. In Appendix A.1 we compare the Bethe equations for the Hubbard chain with boundary fields and the Bethe equations derived in section 3.2, showing that they are equivalent.

## 2 Free fermions with dephasing and boundary losses

Here we consider the fermionic tight-binding chain described by the Hamiltonian

$$H = \sum_{x=1}^{L-1} (c_x^\dagger c_{x+1} + c_{x+1}^\dagger c_x), \tag{1}$$

where $c_x^\dagger, c_x$ are standard fermionic creation and annihilation operators. The system lives on a chain with $L$ sites. We employ open boundary conditions. Our setup is depicted in Fig. 1. The chain undergoes a nonunitary dynamics described by the Lindblad master equation [1]. The state of the system is described by a density matrix $\rho(t)$. Within the framework of Markovian master equations [1], the dynamics of $\rho$ is obtained by solving the Lindblad equation as

$$\frac{d\rho(t)}{dt} := \mathcal{L}(\rho) = -i[H, \rho(t)] + \sum_{x=1}^{L} \sum_\alpha \left( L_{x,\alpha} \rho(t) L_{x,\alpha}^\dagger - \frac{1}{2} \{ L_{x,\alpha}^\dagger L_{x,\alpha}, \rho(t) \} \right), \tag{2}$$

where $\{,\}$ denotes the anticommutator, and $L_{x,\alpha}$ is the so-called Lindblad operator acting at site $x$. In (2), $\mathcal{L}$ is the Liouvillian. The label $\alpha$ encodes the different types of dissipation.

Specifically, we choose $\alpha = 1$ for global dephasing and $\alpha = 2$ for boundary losses. The Lindblad operator for global dephasing reads as

$$L_{x,1} = \sqrt{\gamma} c_x^\dagger c_x, \quad x \in [1, L]. \tag{3}$$

Localized losses at the edges of the chain are described by

$$L_{x,2} = \sqrt{\gamma^-} c_x, \quad x = 1, L, \tag{4}$$

In (3) and (4), $\gamma$ and $\gamma^-$ are the dephasing and loss rates, respectively. By using (2), it is straightforward to obtain the evolution of the fermionic two-point correlation function $G_{x_1,x_2}(t)$ defined as [52]

$$G_{x_1,x_2}(t) := \mathrm{Tr}(\rho(t) c_{x_1}^\dagger c_{x_2}). \tag{5}$$

The dynamics of $G_{x_1,x_2}$ from a generic initial condition $G_{x_1,x_2}(0)$ is obtained by solving the system of equations as [38]

$$\frac{dG_{x_1,x_2}}{dt} := \mathcal{L}^{(2)}(G_{x_1,x_2}) = i(G_{x_1-1,x_2} + G_{x_1+1,x_2} - G_{x_1,x_2-1} - G_{x_1,x_2+1})$$
$$- \gamma G_{x_1,x_2}(1 - \delta_{x_1,x_2}) - \gamma^- G_{x_1,x_2}(\delta_{x_1,1} + \delta_{x_2,1} + \delta_{x_1,L} + \delta_{x_2,L}), \tag{6}$$

where we define the $L^2 \times L^2$ linear super operator $\mathcal{L}^{(2)}$. The superscript in $\mathcal{L}^{(2)}$ is to stress that $\mathcal{L}^{(2)}$ is not the same Liouvillian appearing in (2), which is a $2^L \times 2^L$ matrix and governs the dynamics of the full-system density matrix. In the following we will refer to $\mathcal{L}^{(2)}$ as the Liouvillian, and to $\mathcal{L}$ (cf. (2)) as the full Liouvillian. In (6) we consider the symmetric situation in which the fermions are removed at the edges of the chain at the same rate $\gamma^-$. However, the case with different rates $\gamma^-_{L(R)}$ at the left and right edges of the chain can be considered as well. The first term in (6) describes the unitary dynamics governed by the Hamiltonian (1). The second term is the dephasing, which suppresses the off-diagonal elements of $G_{x_1,x_2}$. The last term describes incoherent absorption of fermions at the edges of the chain.

Importantly, the solution of (6) allows one to obtain the dynamics of $G_{x_1,x_2}$ in several physical situations. For instance, let us consider the setup investigated in Ref. [45], in which a free-fermion chain is subject to global dephasing and fermion pumping at the left edge of the chain and fermion losses at the right one. Let us focus on the case with dephasing rate $\gamma$, and equal pump/loss rate $\gamma'$. The evolution of $G_{x_1,x_2}$ is obtained by solving [46]

$$\frac{dG_{x_1,x_2}}{dt} := \mathcal{L}^{(2)}(G_{x_1,x_2}) = i(G_{x_1-1,x_2} + G_{x_1+1,x_2} - G_{x_1,x_2-1} - G_{x_1,x_2+1})$$
$$- \gamma G_{x_1,x_2}(1 - \delta_{x_1,x_2}) - \frac{\gamma'}{2} G_{x_1,x_2}(\delta_{x_1,1} + \delta_{x_2,1} + \delta_{x_1,L} + \delta_{x_2,L}) + \gamma' \delta_{x_1,1}\delta_{x_2,1}. \tag{7}$$

Eq. (7) is the same as (6) apart for the boundary terms. The boundary dissipation is modeled by the Lindblad operators $L_{x,2} = \sqrt{\gamma'} c_1^\dagger$ and $L_{x,2} = \sqrt{\gamma'} c_L$. Importantly, Eq. (7) becomes the same as (6) after the redefinition $\gamma^- = \gamma'/2$, apart for the "driving" term $\gamma' \delta_{x_1,1}\delta_{x_2,1}$. However, since Eq. (6) and (7) are linear in $G_{x_1,x_2}$, given the general solution of (6), it is possible to construct the solution of (7) with generic initial condition $G_{x_1,x_2}^{(\mathrm{in})}$. Indeed, let us consider $G_{x_1,x_2}^{(\mathrm{I})}$ solution of (7) without the last term, and with initial condition $G_{x_1,x_2}(0) = G_{x_1,x_2}^{(\mathrm{in})}$. Let us also consider the solution $G_{x_1,x_2}^{(\mathrm{II})}$ of (7) without the driving term and with delta initial condition $G_{x_1,x_2}^{(\mathrm{II})}(0) = \delta_{x_1,1}\delta_{x_2,1}$. Now, one can verify that the solution of (7) is

$$G_{x_1,x_2}(t) = G_{x_1,x_2}^{(\mathrm{I})} + \gamma' \int_0^t d\tau\, G_{x_1,x_2}^{(\mathrm{II})}(t - \tau). \tag{8}$$

In the following sections we will determine the full spectrum, i.e., the eigenvalues and the eigenvectors of $\mathcal{L}^{(2)}$ (cf. (6)), for arbitrary $\gamma, \gamma^-$ by using the Bethe Ansatz. This allows us to derive compact formulas for the fermionic correlator $G_{x_1,x_2}$ at arbitrary long times and chain sizes. In principle, by using (8) this also allows to obtain the dynamics of $G_{x_1,x_2}$ for the setup of Ref. [45].

## 3  Bethe Ansatz treatment of the Liouvillian $\mathcal{L}^{(2)}$

Here we discuss the Bethe Ansatz framework that allows to solve (6). We first introduce the Ansatz *à la* Bethe for the right eigenvectors of $\mathcal{L}^{(2)}$ in section 3.1. In section 3.2 we discuss the solutions of the Bethe equations and the general structure of the eigenvalues (energies) of the Liouvillian $\mathcal{L}^{(2)}$. In section 3.3 we focus on a special class of states, which do not depend on dephasing, i.e., they are the same as in the tight-binding chain with boundary losses. In section 3.4 we discuss eigenvalues of $\mathcal{L}^{(2)}$ that are perturbatively related to the ones of section 3.3 in the large $L$ limit. Finally, in section 3.5 we discuss solutions of the Bethe equations that form perfect strings in the complex plane (see Fig. 2). These states correspond to real eigenvalues and govern the long-time dynamics of the fermion correlator.

### 3.1  Bethe Ansatz for the eigenstates of $\mathcal{L}^{(2)}$

Inspired by the coordinate Bethe Ansatz solution of the $XXZ$ chain with open boundary conditions [53] and by the Bethe Ansatz treatment of dephasing [11,26,27] and incoherent hopping [28] in free-fermion systems, we employ the following Ansatz for $G_{x_1,x_2}$ as

$$
\begin{aligned}
G_{x_1,x_2} = \sum_{r_1,r_2=\pm} r_1 r_2 e^{\varepsilon(k_1,k_2)t} \Big\{ &\Big[ A_{12}(r_1,r_2) e^{ir_1 k_1 x_1 + ir_2 k_2 x_2} \\
&+ (-1)^{x_1+x_2} A_{21}(r_1,r_2) e^{ir_2 k_1 x_2 + ir_1 k_2 x_1} \Big] \Theta(x_2-x_1) \\
&+ \sigma(-1)^{x_1+x_2} \Big[ A_{12}(r_1,r_2) e^{ir_1 k_1 x_2 + ir_2 k_2 x_1} \\
&+ (-1)^{x_1+x_2} A_{21}(r_1,r_2) e^{ir_2 k_1 x_1 + ir_1 k_2 x_2} \Big] \Theta(x_1-x_2) \Big\}.
\end{aligned} \tag{9}
$$

Here $k_1, k_2$ are complex quasimomenta, which have to be determined by solving the so-called Bethe equations. $G_{x_1,x_2}$ (after vectorization) are the right eigenvectors of $\mathcal{L}^{(2)}$ with eigenvalues $\varepsilon(k_1,k_2)$. The evolution of (9) is "simple" because $\mathcal{L}^{(2)}(G_{x_1,x_2}) = \varepsilon G_{x_1,x_2}$, although it is not trivial, due to the eigenvalues $\varepsilon$ being complex. The sum over $r_1, r_2$ in (9) is over the reflections of $k_1, k_2$, similar to the Bethe Ansatz solution of the Heisenberg chain [53] with boundary fields. The functions $\Theta(x)$ are Heaviside step functions. To recover the result for $x_1 = x_2$, one has to take the limit $x_2 = x_1 + \epsilon$, sending $\epsilon > 0$ to zero. This means that $G_{x_1,x_1}$ is given by the prefactor of the first Heaviside function in (9). The coefficients $A_{12}$ and $A_{21}$ are scattering amplitudes, which depend on $k_1, k_2$. Crucially, the Liouvillian $\mathcal{L}^{(2)}$ is invariant under the transformation $\mathcal{R}$ that transforms $G_{x_1,x_2} \to (-1)^{x_1+x_2} G_{x_2,x_1}$ as it can be verified by substitution in (6). Since $\mathcal{R}^2$ is the identity, one has for the eigenfunctions of $\mathcal{R}$ that $(-1)^{x_1+x_2} G_{x_2,x_1} = \sigma G_{x_1,x_2}$, where $\sigma = \pm 1$. The second term in (9) takes into account this symmetry.

To proceed, let us observe that in the bulk of the chain, i.e., for $1 < x_1, x_2 < L$, after substituting (9) in (6), we obtain the condition

$$
i(G_{x_1-1,x_2} + G_{x_1+1,x_2} - G_{x_1,x_2-1} - G_{x_1,x_2+1}) - \gamma(1-\delta_{x_1,x_2})G_{x_1,x_2} - \varepsilon G_{x_1,x_2} = 0. \tag{10}
$$

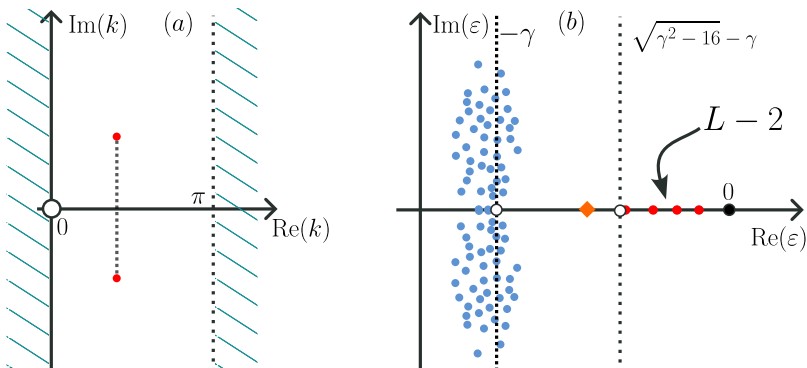

Figure 2: Free-fermion chain with global dephasing and boundary losses. (a) Allowed values for the solutions $k_1$ and $k_2$ of the Bethe equations (cf. (18) and (19)). Only the region $0 < \mathrm{Re}(k) < \pi$ is allowed. Some of the solutions form complex conjugate pairs as $k_1 = k_2^*$, corresponding to real eigenvalues $\varepsilon$. The remaining solutions are all complex but do not form perfect strings. (b) Typical structure of the Liouvillian spectrum: $\mathrm{Im}(\varepsilon)$ versus $\mathrm{Re}(\varepsilon)$. We only consider the situation with $\gamma > 4$. The red circles are purely real eigenvalues. They form an isolated diffusive band of eigenvalues. The diffusive band extends up to $\varepsilon = \sqrt{\gamma^2 - 16} - \gamma$, and it contains $L$ eigenvalues for generic $\gamma > 4$ and $\gamma^-$. Upon lowering $\gamma$ the diffusive band is depleted. The cluster around $\varepsilon = -\gamma$ contains $\sim L(L-1)$ eigenvalues. $L(L-1)/2$ of the levels are trivially obtained from the spectrum of the model at $\gamma = 0$. The diamond denotes a pair of almost degenerate eigenvalues, which correspond to boundary-localized modes of the Liouvillian.

Let us consider the situation with $x_1 \neq x_2$. One can verify that the Ansatz (9) satisfies (10) if we fix

$$\varepsilon(k_1, k_2) = 2i\cos(k_1) - 2i\cos(k_2) - \gamma. \tag{11}$$

Importantly, the minus sign in the second term in (11) depends on the choice of the Ansatz (9). By redefining $k_2 \to k_2 + \pi$ in the terms that contain the sign factor $(-1)^{x_1+x_2}$ in (9), one obtains that $\varepsilon = 2i\cos(k_1) + 2i\cos(k_2) - \gamma$, which is symmetric under exchange $k_1 \leftrightarrow k_2$ (see, for instance, Ref. [11]). Notice that one has to change $k_2 \to k_2 + \pi$ also in the Bethe equations (see section 3.2). After these redefinitions, the new Bethe equations become the same as the Bethe equations for the Hubbard chain with imaginary boundary magnetic fields (see Appendix A.1). This happens despite the fact that the full Liouvillian contains a boundary pair production term (see Appendix A).

Let us now determine the coefficients $A_{12}$ and $A_{21}$ in (9). It is convenient to to treat the cases $\sigma = 1$ and $\sigma = -1$ separately. Let us start with the case with $\sigma = 1$ in (9). As it will be clear in section 3.3, for $\sigma = -1$ the Ansatz (9) does not depend on $\gamma$, and the eigenstates of $\mathcal{L}^{(2)}$ are the same as those of the chain with boundary losses and no bulk dephasing. Let us now impose the "contact" condition obtained by fixing $x_1 = x_2$ in (9) and requiring that Eq. (10) holds with $\varepsilon$ as in (11). A long calculation gives

$$A_{21}(r_1, r_2) = -A_{12}(r_2, r_1) \frac{\gamma/2 + r_2 \sin(k_1) + r_1 \sin(k_2)}{\gamma/2 - r_2 \sin(k_1) - r_1 \sin(k_2)}. \tag{12}$$

Finally, we impose the boundary conditions. For Eq. (10) to be compatible with the Lindblad equation (6) at the boundaries, we require that

$$iG_{0,x_2} + \gamma^- G_{1,x_2} = 0, \tag{13}$$

$$iG_{x_1,L+1} - \gamma^- G_{x_1,L} = 0. \tag{14}$$

The conditions (13) and (14) give eight equations. They allow us to fix

$$A_{12}(-1,1) = \frac{1 - ie^{ik_1}\gamma^-}{1 - ie^{-ik_1}\gamma^-} A_{12}(1,1), \tag{15}$$

$$A_{12}(1,-1) = e^{2ik_2(1+L)}\frac{1 + ie^{-ik_2}\gamma^-}{1 + ie^{ik_2}\gamma^-} A_{12}(1,1), \tag{16}$$

$$A_{12}(-1,-1) = e^{2ik_2(1+L)}\frac{1 - ie^{ik_1}\gamma^-}{1 - ie^{-ik_1}\gamma^-}\frac{1 + ie^{-ik_2}\gamma^-}{1 + ie^{ik_2}\gamma^-} A_{12}(1,1). \tag{17}$$

Moreover, one obtains two more equations (Bethe equations) that provide the quantization conditions for $k_1$ and $k_2$. Before discussing the Bethe equations, let us stress that it is natural to expect that Eq. (13) and (14) can be modified to account for different loss rates $\gamma_L^-$ and $\gamma_R^-$ at the two edges of the chain.

## 3.2 Bethe equations and general structure of the Liouvillian spectrum

The two extra conditions obtained from (13) and (14) provide two coupled nonlinear equations for $k_1, k_2$ as

$$e^{2ik_1(L-1)}\left(\frac{e^{ik_1} - i\gamma^-}{e^{-ik_1} - i\gamma^-}\right)^2 = \prod_{r_2 = \pm 1}\frac{\gamma/2 - \sin(k_1) + r_2\sin(k_2)}{\gamma/2 + \sin(k_1) - r_2\sin(k_2)}, \tag{18}$$

$$e^{2ik_2(L-1)}\left(\frac{e^{ik_2} + i\gamma^-}{e^{-ik_2} + i\gamma^-}\right)^2 = \prod_{r_1 = \pm 1}\frac{\gamma/2 - r_1\sin(k_1) + \sin(k_2)}{\gamma/2 + r_1\sin(k_1) - \sin(k_2)}. \tag{19}$$

Eq. (18) and (19) differ from the Bethe equations for the periodic tight-binding chain with dephasing. For periodic boundary conditions one has that $\gamma^- = 0$ and one has to replace $e^{2ik_jL} \rightarrow e^{ik_jL}$. Moreover, only one of the two terms in the right-hand side survives, because there is no product over the reflections of the quasimomenta.

Let us now discuss some properties of the Bethe equations (18) and (19). The total number of solutions is $L^2$ because the Liouvillian $\mathcal{L}^{(2)}$ is a $L^2 \times L^2$ matrix. The momenta $k_1, k_2$ are all complex. The allowed domain of $k_j$ is reported in Fig. 2 (a), plotting $\text{Im}(k_j)$ versus $\text{Re}(k_j)$. The Bethe equations possess several symmetries that we now discuss. Given a generic pair of quasimomenta $(k_1, k_2)$ solving (18) and (19), the pairs obtained by arbitrary reflections $\pm k_1$ and $\pm k_2$ are also solutions of (18) and (19). This can be used to fix $\text{Re}(k_j) > 0$. Notice that $k_j = 0$ and $k_j = \pi$ are solutions of the Bethe equations, although they have to be discarded because they give vanishing eigenvectors (9). The invariance of (18) and (19) under $k_1 \rightarrow \pm k_2 \pm \pi$ can be exploited to fix $\text{Re}(k_j) < \pi$. Since the imaginary part of $k_j$ can be arbitrary, the solutions of the Bethe equations live in the strip $(0, \pi) \times (-i\infty, i\infty)$ (see Fig. 2 (a)). Another important property of (18) and (19) is that given a pair $(k_1, k_2)$ solving (18) (19), then $(k_2^*, k_1^*)$, with the star denoting complex conjugation, is also a solution. This means that (cf. (11)) the eigenvalues $\varepsilon$ appear in complex conjugated pairs.

Crucially, some of the solutions $(k_1, k_2)$ form complex conjugate pairs (see Fig. 2 (a)), i.e., $k_1 = k_2^*$. These solutions form "strings" patterns in the complex plane (see Fig. 2). The corresponding eigenvalues are real. We anticipate that these solutions will determine the behavior of the fermionic correlator at long times, because the solutions giving the eigenvalues with the larger real parts will be of this type. String solutions of the Bethe equations can be effectively described in the limit $L \rightarrow \infty$ by using the framework of the string hypothesis [54]. As we will discuss in section 3.5, in the large $L$ limit the imaginary part of $k_1, k_2$ can be derived by solving a nonlinear equation, similar to the so-called Bethe-Gaudin-Takahashi (BGT) equation that appears within the framework of the string hypothesis for integrable models [54]. Corrections to the string hypothesis are exponentially suppressed in the limit $L \rightarrow \infty$.

Let us now discuss the general structure of the spectrum of $\mathcal{L}^{(2)}$. This is illustrated in Fig. 2 (b). First, for $\gamma^- = 0$, there is a zero-energy state $\varepsilon = 0$, which corresponds to the steady-state of the system at $t \to \infty$. Within the Bethe Ansatz treatment, the steady state corresponds to diverging momenta $k_1, k_2$. At nonzero $\gamma^-$, the eigenvalue $\varepsilon = 0$ disappears. We can generically distinguish two different regions in the energy spectrum. The eigenvalues $\varepsilon$ with the larger real parts form a band of *real* energy near $\varepsilon = 0$. As these solutions are responsible for diffusive behavior [26, 28], we dub them diffusive band [26]. These are reminiscent of the diffusive band appearing for the periodic chain with dephasing [26], or incoherent hopping [28]. The number of eigenvalues in the band depends on $\gamma$. As it will be clear in the following, for $\gamma > 4$, the diffusive band contains $L$ eigenvalues at small $\gamma^-$, which is the largest possible number of states. Interestingly, for $\gamma^- > \gamma_c^- = \exp(-\mathrm{arccosh}(\gamma/4))$ (see section 3.5) two of the solutions move outside of the diffusive band towards lower $\mathrm{Re}(\varepsilon)$ (see the diamond symbol in Fig. 2). Concomitantly, the two eigenvalues become almost degenerate. Precisely, their splitting in energy decays exponentially with $L$. These states are boundary-related, and are present also in the one-dimensional Hubbard model with boundary fields [25]. Boundary-related states have been investigated in the two-particle sector for the open $XXZ$ spin chain in Ref. [55].

Furthermore, a cluster of eigenvalues is present around $\varepsilon = -\gamma$. As it will be clear in section 3.3, there are $L(L-1)/2$ eigenvalues that are related by a $\gamma$ shift to the eigenvalues of $\mathcal{L}^{(2)}$ with $\gamma = 0$, i.e., with only losses. Specifically, they are given by (9) with $\sigma = -1$. The associated Bethe equations are decoupled and are given by (24) and (25). The remaining complex eigenvalues correspond states with $\sigma = 1$ in (9). Still, in the large $L$ limit they differ from the states with $\sigma = -1$ by $\mathcal{O}(1/L)$ terms, i.e., they are "perturbatively" related to the states with $\sigma = -1$.

### 3.3 Dephasing-independent solutions

Let us now characterize the states (9) with $\sigma = -1$. Now, the main difference with the case $\sigma = 1$ is that the contact condition (12) has to be modified as

$$A_{21}(r_1, r_2) = -A_{12}(r_2, r_1). \tag{20}$$

The boundary conditions to be imposed are the same as (13) and (14). They give

$$A_{12}(-1,1) = \frac{1 - ie^{ik_1}\gamma^-}{1 - ie^{-ik_1}\gamma^-} A_{12}(1,1), \tag{21}$$

$$A_{12}(1,-1) = \frac{1 + ie^{ik_2}\gamma^-}{1 + ie^{-ik_2}\gamma^-} A_{12}(1,1), \tag{22}$$

$$A_{12}(-1,-1) = \frac{1 - ie^{ik_1}\gamma^-}{1 - ie^{-ik_1}\gamma^-} \frac{1 + ie^{ik_2}\gamma^-}{1 + ie^{-ik_2}\gamma^-} A_{12}(1,1). \tag{23}$$

Notice that there is no dependence on $L$ in (21)-(23), in contrast with the case with $\sigma = 1$. The Bethe equations now read as

$$(\gamma^-)^2 \sin(k_2(1-L)) + 2i\gamma^- \sin(k_2 L) + \sin(k_2(L+1)) = 0, \tag{24}$$

$$(\gamma^-)^2 \sin(k_1(1-L)) - 2i\gamma^- \sin(k_1 L) + \sin(k_1(L+1)) = 0. \tag{25}$$

The Bethe equations for $k_1$ and $k_2$ are decoupled, reflecting that the system is noninteracting. Also, $k_1, k_2$ do not depend on the dephasing rate $\gamma$. The eigenvalues $\varepsilon$ are the same as in (11), implying that the dependence on $\gamma$ is only a shift. As it is clear from (25), given a solution $k_1$, then $-k_1$ and $k_1 \pm \pi$ is also a solution. The same holds for $k_2$. This means that one can restrict to $0 < \mathrm{Re}(k_j) < \pi$. Moreover, the solutions of Eq. (24) and Eq. (25) are related by complex conjugation.

Eq. (24) and (25) are the same equations describing the tight-binding chain with boundary losses and no bulk dephasing [56]. Since Eq. (24) has $L$ solutions $k_1^{(p)}$ with $p = 1, \ldots, L$, the pairs $(k_1^{(p)}, (k_1^{(q)})^*)$ give all the $L^2$ eigenvalues of the Liouvillian. Upon switching on $\gamma$, only $L(L-1)/2$ survive. These correspond to the pairs $(k_1, k_2)$ such that $k_j \neq 0, \pi$. Moreover, one has to exclude all the pairs $k_1, k_2$ such that $k_1 + k_2 = 0 \mod \pi$, and the pairs $(k_1', k_2')$ that are obtained as $(k_1', k_2') = (\pi - k_2, \pi - k_1)$ from a set of solutions $(k_1, k_2)$. Indeed, one can check that the total number of pairs satisfying these constraints is $L(L-1)/2$. The conditions on $k_1, k_2$ discussed above are the same as in the beginning of Section 3.2. Let us also observe that in the limit $L \to \infty$ the solutions of (24) and (25) are given as

$$k_{1,2} = \frac{\pi}{L+1} j_{1,2} + \mathcal{O}(1/L), \quad j_{1,2} = 1, 2, \ldots L. \tag{26}$$

Specifically, the imaginary part of $k_{1,2}$ is $\mathcal{O}(1/L)$, although it is nonzero. Clearly, Eq. (26) is exact without the $\mathcal{O}(1/L)$ correction for $\gamma^- = 0$. In the last case the eigenvalues $\varepsilon$ form a straight line parallel to the imaginary axis (see Fig. 2), with real part $-\gamma$.

## 3.4 Solutions with vanishing imaginary parts of $k_1$ and $k_2$

Near the eigenvalues that correspond to dephasing-independent solutions of the Bethe equations, there are $\sim L(L-1)/2$ eigenvalues that correspond to $\sigma = 1$ in (9), and that in the large $L$ limit differ by terms $\mathcal{O}(1/L)$ from the dephasing-independent eigenvalues. The number of eigenvalues depends on $\gamma$. In particular, for $\gamma > 4$ their number is exactly $L(L-1)/2$.

We now discuss them restricting to the case with $\gamma^- = 0$. A similar analysis can be performed for nonzero $\gamma^-$. The large $L$ behavior of the Bethe equations (18) (19) suggests the expansion

$$k_1 = k_1^{(r,0)} + k_1^{(r,2)} L^{-2} + i k_1^{(i,1)} L^{-1}, \tag{27}$$

$$k_2 = k_2^{(r,0)} + k_2^{(r,2)} L^{-2} + i k_2^{(i,1)} L^{-1}. \tag{28}$$

Here $k_j^{(r,0)}, k_j^{(r,2)}$ and $k_j^{(i,1)}$ ($j = 1, 2$) are real parameters that have to be determined. After substituting the Ansatz (27) and (28) in the Bethe equations (18) and (19), Taylor expanding in the large $L$ limit, and equating the coefficients of the terms with the same powers of $L$, we obtain

$$2k_1^{(r,0)}(L+1) = j_1 \pi, \quad 2k_2^{(r,0)}(L+1) = j_2 \pi, \quad \text{with} \quad j_1, j_2 = 1, 2, \ldots 2(L+1). \tag{29}$$

We now provide the expression for $k_j^{(i,1)}$. A similar expression can be obtained for $k_j^{(r,2)}$, although since it is cumbersome we do not report it. We obtain

$$k_1^{(i,1)} = -\frac{1}{2} \frac{L}{L+1} \ln \left[ (-1)^{j_1} \frac{\gamma^2 - 2\cos(2k_1^{(r,0)}) + 2\cos(2k_2^{(r,0)}) - 4\gamma \sin(k_1^{(r,0)})}{\gamma^2 - 2\cos(2k_1^{(r,0)}) + 2\cos(2k_2^{(r,0)}) + 4\gamma \sin(k_1^{(r,0)})} \right], \tag{30}$$

$$k_2^{(i,1)} = -\frac{1}{2} \frac{L}{L+1} \ln \left[ (-1)^{j_2} \frac{\gamma^2 + 2\cos(2k_1^{(r,0)}) - 2\cos(2k_2^{(r,0)}) + 4\gamma \sin(k_2^{(r,0)})}{\gamma^2 + 2\cos(2k_1^{(r,0)}) - 2\cos(2k_2^{(r,0)}) - 4\gamma \sin(k_2^{(r,0)})} \right]. \tag{31}$$

Consistency with (27) and (28) requires that $k_j^{(i,1)}$ is real. One can readily check that for $\gamma > 4$ the term inside the square brackets in (30) and (31) is positive provided that $j_1$ and $j_2$ are both even. For $\gamma < 4$, Eq. (30) and (31) are correct only for the eigenvalues near $\varepsilon = -\gamma$. Oppositely, away from $\varepsilon = -\gamma$ the eigenvalues are affected by the presence of the diffusive band, and are not accurately described by (30) and (31). Notice that $k_{1,2}^{(i,1)}$ vanish in the large $L$ limit, even at $j_{1,2}/L$ fixed. The eigenvalues $\varepsilon$ that correspond to solutions of the Bethe

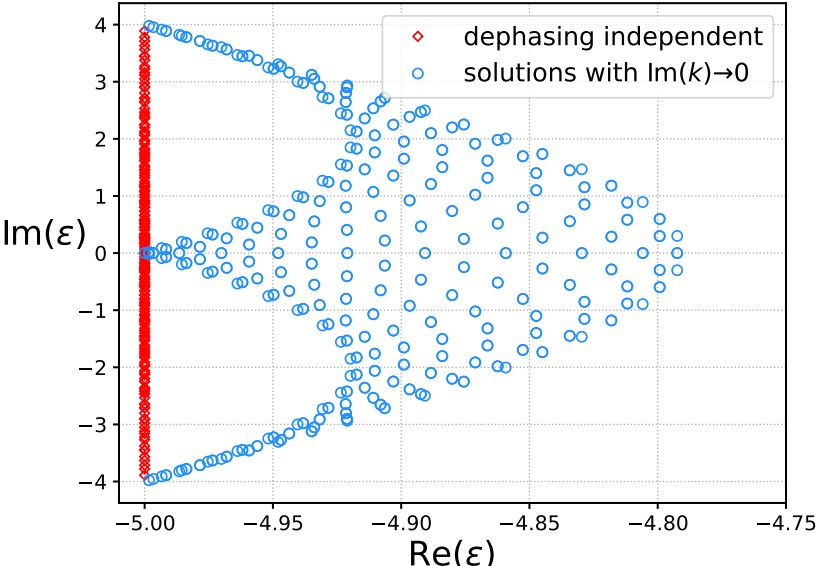

Figure 3: Spectrum of $\mathcal{L}^{(2)}$ for $L = 20$, $\gamma = 5$ and $\gamma^- = 0$. We plot $\text{Im}(\varepsilon)$ versus $\text{Re}(\varepsilon)$. We only show the eigenvalues near $\varepsilon = -\gamma$. The circles correspond to the complex solutions $(k_1, k_2)$ of the Bethe equations with vanishing imaginary parts $\text{Im}(k_j) \to 0$ in the limit $L \to \infty$ (see section 3.4). The diamonds are the solutions that do not depend on the dephasing rate $\gamma$ (see section 3.3).

equations with vanishing imaginary parts are discussed in Fig. 3. We consider only the case with $\gamma = 5$ because for $\gamma > 4$ the diffusive band at smaller $\text{Re}(\varepsilon)$ is well separated from bulk of the spectrum and (27) and (28) are accurate. The diamonds in the Figure correspond to the eigenstates with $\sigma = -1$ in (9) discussed in section 3.3. The circles are the eigenvalues obtained from to momenta of the type (27) and (28). It is interesting to focus on the inner and outer "envelope" of the eigenvalues. They are obtained from (29) (30) and (31) as follows. We checked that the levels $\varepsilon_{\text{in}}$ of the inner envelope correspond to the choice $j_1 = 2$ and $j_2 \in [2, 2(L+1)]$ and $j_1 \in [2, 2(L+1)]$ in (29). The levels $\varepsilon_{\text{out}}$ forming the outer envelope are obtained by choosing $j_1 \in [2, 2(L+1)]$ and $j_2 = 2(L+1) - j_1$.

## 3.5 Diffusive band & boundary states

As we anticipated, the eigenvalues of $\mathcal{L}^{(2)}$ having the largest real part form a diffusive band, and are real. These states correspond to string solutions of the Bethe equations (see Fig. 2 (a)) with nonvanishing imaginary parts in the limit $L \to \infty$. They form complex conjugate pairs $(k_1, k_1^*)$. It is convenient to define $k_1 = k_r + ik_i$ and $k_2 = k_r - ik_i$. We start discussing the case with $\gamma^- = 0$. In terms of $k_r, k_i$, the Bethe equations (18) and (19) become

$$e^{2(L+1)(ik_r - k_i)} = \frac{(\gamma - 4\cosh(k_i)\sin(k_r))(\gamma - 4i\cos(k_r)\sinh(k_i))}{(\gamma + 4\cosh(k_i)\sin(k_r))(\gamma + 4i\cos(k_r)\sinh(k_i))}, \tag{32}$$

$$e^{2(L+1)(ik_r + k_i)} = \frac{(\gamma + 4\cosh(k_i)\sin(k_r))(\gamma - 4i\cos(k_r)\sinh(k_i))}{(\gamma - 4\cosh(k_i)\sin(k_r))(\gamma + 4i\cos(k_r)\sinh(k_i))}. \tag{33}$$

To proceed, we can assume without loss of generality that $k_i > 0$. In the limit $L \to \infty$ the left-hand side of (33) diverges exponentially. This suggests that the denominator in the right-hand side of (32) vanishes. For consistency we can impose that

$$\gamma - 4\cosh(k_i)\sin(k_r) = 0. \tag{34}$$

Similar observations are at the heart of the string hypothesis in Bethe Ansatz solvable models [54]. Solving (34) for $k_r$, we obtain

$$k_r = \pi - \arcsin\left(\frac{\gamma}{4\cosh(k_i)}\right). \tag{35}$$

To remove the singular denominator in (32) we take the product of (32) and (33), and after using (35), we obtain that $k_i$ satisfies the equation

$$\left(\frac{4}{\gamma\,\mathrm{sech}(k_i) - i\sqrt{16 - \gamma^2\mathrm{sech}^2(k_i)}}\right)^{4(L+1)} = \left(\frac{\gamma\,\mathrm{cosech}(k_i) + i\sqrt{16 - \gamma^2\mathrm{sech}^2(k_i)}}{\gamma\,\mathrm{cosech}(k_i) - i\sqrt{16 - \gamma^2\mathrm{sech}^2(k_i)}}\right)^2. \tag{36}$$

The derivation of (36) is similar to that of the Bethe-Gaudin-Takahashi (BGT) equations for the Hubbard chain [25, 54]. For this reason, we refer to (36) as the BGT equation. The derivation can be extended to the case with nonzero $\gamma^-$. The BGT equation becomes

$$\frac{(4e^{k_i} + \gamma\gamma^-\mathrm{sech}(k_i) + i\gamma^-\mathcal{B}(k_i))^2(4e^{-k_i} - \gamma\gamma^-\mathrm{sech}(k_i) - i\gamma^-\mathcal{B}(k_i))^2}{(4e^{k_i}\gamma^- - \gamma\,\mathrm{sech}(k_i) - i\mathcal{B}(k_i))^2(4e^{-k_i}\gamma^- + \gamma\,\mathrm{sech}(k_i) + i\mathcal{B}(k_i))^2}$$

$$\times \frac{(\gamma\,\mathrm{cosech}(k_i) + i\mathcal{B}(k_i))^2}{(\gamma\,\mathrm{cosech}(k_i) - i\mathcal{B}(k_i))^2} - \left(\frac{4}{\gamma\,\mathrm{sech}(k_i) - i\mathcal{B}(k_i)}\right)^{4L} = 0, \tag{37}$$

where we defined

$$\mathcal{B}(x) := \sqrt{16 - \gamma^2\mathrm{sech}^2(x)}. \tag{38}$$

After solving (37) for $k_i$, we obtain $k_r$ by using (35). It is convenient to take the logarithm of both terms in (37) to obtain the BGT equations in logarithmic form. Let us first define

$$z := \frac{\gamma}{4\cosh(k_i)}, \quad k_i = \mathrm{arccosh}\left(\frac{\gamma}{4z}\right). \tag{39}$$

After taking the logarithm of both members in (37) and using (39), we obtain

$$2iL\arcsin(z_j) + \ln\left[\frac{z_j\gamma + i\sqrt{1 - z_j^2}\sqrt{\gamma^2 - 16z_j^2}}{z\gamma - i\sqrt{1 - z_j^2}\sqrt{\gamma^2 - 16z_j^2}}\right]$$

$$+ \ln\left[\frac{\gamma^-(-iz_j + \sqrt{1 - z_j^2})(\gamma + \sqrt{\gamma^2 - 16z_j^2}) + 4iz_j}{(iz_j - \sqrt{1 - z_j^2})(\gamma + \sqrt{\gamma^2 - 16z_j^2}) + 4iz_j\gamma^-}\right]$$

$$+ \ln\left[\frac{\gamma + \sqrt{\gamma^2 - 16z_j^2} + 4z_j(z_j + i\sqrt{1 - z_j^2})\gamma^-}{4z_j(z_j + i\sqrt{1 - z_j^2}) - (\gamma + \sqrt{\gamma^2 - 16z_j^2})\gamma^-}\right] = -\pi i I_j. \tag{40}$$

Here $I_j \in [0, L-2)$ are integers, forming the so-called BGT quantum numbers, which identify the different solutions $z_j$. They originate from the branch cut of the logarithm. The energy with the largest real part corresponds to $I_j = 0$. Notice that here we assume $\gamma^- > 0$. For $\gamma^- = 0$ $I_j = 0$ has to be excluded, because it would correspond to $\varepsilon = 0$. The number of solutions in the diffusive band depends on $\gamma$ and $\gamma^-$. Specifically, for $\gamma > 4$ there are at least $L - 2$ solutions of (40). Two extra solutions appear provided that $\gamma^-$ is small enough. Precisely, for $\gamma^- > \gamma_c^-$ two of the eigenvalues detach from the diffusive band, moving towards lower $\mathrm{Re}(\varepsilon)$. They correspond to boundary-related modes of the Liouvillian $\mathcal{L}^{(2)}$ (see diamond symbol in Fig. 2). Similar boundary states appear in the open Hubbard chain with boundary magnetic

fields [25]. Precisely, from (37) we obtain that the boundary-related states are present for $\gamma^- > \gamma_c^-$ given as

$$\gamma_c^- = \exp\left(-\operatorname{arccosh}(\gamma/4)\right). \tag{41}$$

The condition (41) is obtained by noticing that at the left edge of the diffusive band (see Fig. 2) one has that $k_i = \operatorname{arccosh}(\gamma/4)$, and by solving Eq. (37) for $\gamma^-$. Eq. (41) holds true for $\gamma > 4$, and, since we are employing the framework of the string hypothesis, in the thermodynamic limit $L \to \infty$. Finally, we should stress that to extract the quasimomenta that correspond to the boundary-related eigenvalues it is convenient to use Eq. (37) rather than the logarithmic BGT equation (40), as it will be clear in section 5.2 (see Fig. 6).

Let us now discuss the gap of the Liouvillian $\mathcal{L}^{(2)}$. This is obtained by considering the energy $\varepsilon$ with the largest nonzero real part. Precisely, the gap $\Delta\mathcal{L}^{(2)}$ is defined as

$$\Delta\mathcal{L}^{(2)} := -\max_j \operatorname{Re}(\varepsilon_j), \quad \text{with } \operatorname{Re}(\varepsilon_j) \neq 0. \tag{42}$$

We numerically verified that for nonzero $\gamma^-$, $\Delta\mathcal{L}^{(2)}$ corresponds to $I_j = 0$ in (40). For $\gamma^- = 0$ one has to choose $I_j = 1$. Focusing on $\gamma^- \neq 0$, a straightforward expansion of (40) for $I_j = 0$ in the large $L$ limit gives

$$z \simeq \frac{\pi}{2}\frac{1}{L} + \left(\frac{\pi}{2} - \frac{\pi(1-(\gamma^-)^2)}{\gamma\gamma^-}\right)\frac{1}{L^2}, \tag{43}$$

where we neglected higher order terms in $1/L$. After substituting in the expression for the energy (11) we obtain the gap of the Liouvillian as

$$\Delta\mathcal{L}^{(2)} \simeq -\frac{2\pi^2}{\gamma L^2} + \frac{4\pi^2(2-\gamma\gamma^- + 2(\gamma^-)^2)}{\gamma^2\gamma^-}\frac{1}{L^3}. \tag{44}$$

At the leading order in $1/L$, the gap depends only on the bulk dephasing. At higher orders a dependence on $\gamma^-$ appears. This reflects that the effect of the boundaries appears at later times.

## 4 Asymptotic dynamics of the fermionic two-point function

Here we derive a formula for the time-dependent correlation function $G_{x_1,x_2}(t)$ starting from an arbitrary initial condition $G_{x_1,x_2}(0)$. The strategy is to build a complete basis of operators by using the Bethe states (9). This basis is then used to expand the initial condition for the correlator. In section 4.1 we construct the complete basis for the generic two-point correlation function, using the left and right eigenvectors of $\mathcal{L}^{(2)}$. In section 4.2 we compute the leading contribution of the norm of the Bethe states (9) in the large $L$ limit. We only consider the states forming the diffusive band, because they are dominant in the long-time limit. Finally, in section 4.3 we derive the long-time limit of the density profile, i.e., the diagonal correlator $G_{x,x}(t)$.

### 4.1 Left and right eigenvectors of $\mathcal{L}^{(2)}$

One can decompose the initial correlator $G_{x_1,x_2}(0)$ in the basis of eigenvectors of $\mathcal{L}^{(2)}$. Let us denote with $G_{x_1,x_2}^{(k_1,k_2)}$ the eigenvector of $\mathcal{L}^{(2)}$ identified by the solutions $k_1, k_2$ of the Bethe equations (18) (19). The dynamics of $G_{x_1,x_2}^{(k_1,k_2)}$ is given as

$$\frac{dG_{x_1,x_2}^{(k_1,k_2)}}{dt} = \mathcal{L}^{(2)}(G_{x_1,x_2}^{(k_1,k_2)}) = \varepsilon(k_1,k_2)G_{x_1,x_2}^{(k_1,k_2)}. \tag{45}$$

The generic correlator $G_{x_1,x_2}$ can be decomposed as

$$G_{x_1,x_2}(t) = \sum_{\{k_1,k_2\}} |k_1,k_2\rangle\langle k_1,k_2|0\rangle e^{\varepsilon(k_1,k_2)t}, \quad |k_1,k_2\rangle := N_{k_1,k_2}^{-1} G_{x_1,x_2}^{(k_1,k_2)}. \tag{46}$$

where the sum is over the solutions of the Bethe equations $\{k_1,k_2\}$ (cf. (18) and (19)). In (46) we redefined $|0\rangle := G_{x_1,x_2}(0)$, and we defined $\langle k_1,k_2| := \bar{G}_{x_1,x_2}^{(k_1,k_2)}$ as the left eigenvector of the Liouvillian. Since the Liouvillian is not hermitian we have that $\langle k_1,k_2| \neq (|k_1,k_2\rangle)^\dagger$. In (46) we used the "scalar product"

$$\langle k_1,k_2|k_3,k_4\rangle := \sum_{z_1,z_2=1}^{L} \bar{G}_{z_1,z_2}^{(k_1,k_2)} G_{z_1,z_2}^{(k_3,k_4)}. \tag{47}$$

Similar definition holds for the scalar product with the initial correlator $|0\rangle$, i.e., $\langle k_1,k_2|0\rangle$. In (46) $N_{k_1,k_2} := \langle k_1,k_2|k_1,k_2\rangle$ is the normalization of the eigenvectors. Following Ref. [28], it is possible to determine $\langle k_1,k_2|$ by observing that the energy (11) is invariant under the redefinition

$$k_1 = \pi - k_2, \quad k_2 = \pi - k_1. \tag{48}$$

This leads to the definition

$$\langle k_1,k_2| := |\pi - k_2, \pi - k_1\rangle. \tag{49}$$

The Ansatz (9) gets modified. Apart from the redefinition (48), one obtains new scattering amplitudes $\bar{A}_{12}$ and $\bar{A}_{21}$. For $\sigma = 1$ they read as

$$\bar{A}_{21}(r_1,r_2) = -\bar{A}_{12}(r_2,r_1) \frac{\gamma/2 + r_1 \sin(k_1) + r_2 \sin(k_2)}{\gamma/2 - r_1 \sin(k_1) - r_2 \sin(k_2)}. \tag{50}$$

Notice that (50) differs from (12) by the exchange $r_1 \leftrightarrow r_2$ in the ratio on the right-hand side. Moreover, we have that

$$\bar{A}_{12}(-1,1) = \frac{1 + i e^{-ik_2}\gamma^-}{1 + i e^{ik_2}\gamma^-} \bar{A}_{12}(1,1), \tag{51}$$

$$\bar{A}_{12}(1,-1) = e^{-2ik_1(L+1)} \frac{1 - i e^{ik_1}\gamma^-}{1 - i e^{-ik_1}\gamma^-} A_{12}(1,1), \tag{52}$$

$$\bar{A}_{12}(-1,-1) = e^{-2ik_1(L+1)} \frac{1 - i e^{ik_1}\gamma^-}{1 - i e^{-ik_1}\gamma^-}, \frac{1 + i e^{-ik_2}\gamma^-}{1 + i e^{ik_2}\gamma^-} \bar{A}_{12}(1,1). \tag{53}$$

Importantly, the Bethe equations remain the same as in (18) and (19).

Let us now discuss the case of $\sigma = -1$. Repeating the steps as above we obtain that

$$\bar{A}_{21}(r_1,r_2) = -\bar{A}_{12}(r_2,r_1). \tag{54}$$

Moreover, we have

$$\bar{A}_{12}(-1,1) = \frac{1 + i e^{-ik_2}\gamma^-}{1 + i e^{ik_2}\gamma^-} \bar{A}_{12}(1,1), \tag{55}$$

$$\bar{A}_{12}(1,-1) = \frac{1 - i e^{-ik_1}\gamma^-}{1 - i e^{ik_1}\gamma^-} \bar{A}_{12}(1,1), \tag{56}$$

$$\bar{A}_{12}(-1,-1) = \bar{A}_{12}(1,-1)\bar{A}_{12}(-1,1)/\bar{A}_{12}(1,1). \tag{57}$$

We anticipate, however, that since the states with $\sigma = -1$ have typically small $\text{Re}(\varepsilon)$, they do not contribute significantly at long times. Moreover, for delta initial conditions $G_{x_1,x_2}(0)$ considered in section 4.3 their contribution is exactly zero.

## 4.2 Norm of the eigenvectors

Here we derive the norm $N_{k_1,k_2}$ for a Bethe eigenstate (9) characterized by generic solutions of the Bethe equations $k_1, k_2$. To proceed, let us focus on eigenstates with $\sigma = 1$ (cf. (9)) because they dominate the dynamics at long times. By using (9) (47) and (49) we obtain that

$$N_{k_1,k_2} = \langle k_1, k_2 | k_1, k_2 \rangle = N_0 + N_1 L + N_2 L^2 \,, \tag{58}$$

where $N_0, N_1, N_2$ are functions of $k_1, k_2$. A tedious calculation gives

$$N_2 = -8 \frac{\gamma/2 + \sin(k_1) + \sin(k_2)}{\gamma/2 - \sin(k_1) - \sin(k_2)} \,. \tag{59}$$

A similar calculation gives $N_1$ as

$$
\begin{aligned}
N_1 = 4i\gamma(\cos(k_1) - \cos(k_2)) & \left[ \frac{1}{(\gamma/2 - \sin(k_1) - \sin(k_2))^2} \right. \\
& + \frac{\gamma/2 + \sin(k_1) + \sin(k_2)}{(\gamma/2 - \sin(k_1) - \sin(k_2))(\gamma/2 - \sin(k_1) + \sin(k_2))(\gamma/2 + \sin(k_1) - \sin(k_2))} \Bigg] \\
& - \frac{\gamma/2 + \sin(k_1) + \sin(k_2)}{\gamma/2 - \sin(k_1) - \sin(k_2)} \left[ \frac{8(1 + (\gamma^-)^2)}{(1 - i\gamma^- e^{ik_1})(1 - i\gamma^- e^{-ik_1})} \right. \\
& \left. + \frac{8(1 + (\gamma^-)^2)}{(1 + i\gamma^- e^{ik_2})(1 + i\gamma^- e^{-ik_2})} \right].
\end{aligned} \tag{60}
$$

Finally, we observe that $N_0$ is in general nonzero. This is in contrast with the case of the tight-binding chain with incoherent hopping and periodic boundary conditions [28]. Crucially, we notice that for the string solutions of the Bethe equations forming the diffusive band (see section 3.5) both $N_1$ and $N_2$ are singular in the limit $L \to \infty$ because $\gamma/2 - \sin(k_1) - \sin(k_2)$ vanishes. Similar divergences will plague the overlaps $\langle k_1, k_2 | 0 \rangle$ and $| k_1, k_2 \rangle$. This means that to extract the dynamics of $G_{x_1,x_2}$ one has to take carefully the limit $L \to \infty$, i.e., going beyond the string hypothesis. In the following section we will show that to obtain the leading behavior of the norm in the large $L$ limit it is sufficient to consider the term in (60).

## 4.3 Evolution of the density profile

In this section we provide analytic results for the dynamics of $G_{x,x}(t)$ starting from the initial condition

$$|0\rangle := G_{x_1,x_2}(0) = \delta_{x_1,x} \delta_{x_2,x} \,, \tag{61}$$

which corresponds to a fermion initially localized at position $x$. Our results hold in the long-time limit and for large $L$.

Crucially, since Eq. (6) is linear as a function of $G_{x_1,x_2}$, its solution with the delta initial condition (61) is sufficient to obtain the dynamics of $G_{x_1,x_2}$ starting from an arbitrary initial density profile. Precisely, let us consider a generic diagonal initial condition as

$$|0\rangle = \delta_{x_1,x_2} f(x_2), \tag{62}$$

where $0 \leq f(x) \leq 1$. Given the solution $G^{(x)}_{x_1,x_2}(t)$ with initial condition (61) at fixed $x$, the solution $G_{x_1,x_2}(t)$ with initial condition (62) is obtained as

$$G_{x_1,x_2}(t) = \sum_{x=1}^{L} G^{(x)}_{x_1,x_2}(t) f(x). \tag{63}$$

Let us now discuss the time-dependent correlator $G_{x_1,x_2}(t)$ starting from (61). To be specfic, let us consider the situation with a fermion initially localized at $x$ away from the boundaries, i.e., with $x/L \neq 0,1$ in (61). To obtain $G_{x_1,x_2}(t)$ we employ (46), restricting ourselves to the eigenstates of the Liouvillian forming the diffusive band (see section 3.5). Moreover, we exploit the string hypothesis, which holds in the limit $L \to \infty$. A straightforward calculation gives the overlaps between (61) and the generic Bethe eigenstate (9) as

$$\langle k_1, k_2 | 0 \rangle = \sum_{r_1, r_2 = \pm} r_1 r_2 \left[ e^{-i(r_2 k_1 + r_1 k_2)x} \bar{A}_{12}(r_1, r_2) + e^{-i(r_1 k_1 + r_2 k_2)x} \bar{A}_{21}(r_1, r_2) \right], \tag{64}$$

where we used (49) and (48). Eq. (64) is valid for all the Bethe eigenstates (9). However, it is straightforward to check that the eigenstates with $\sigma = -1$ (see section (3.3)) have zero overlap with (61).

Let us now restrict ourselves to the eigenstates forming the diffusive band (see Fig. 2 (b)). As discussed in section 3.5, the corresponding solutions of the Bethe equations form complex conjugated pairs, and can be treated by means of the string hypothesis. As anticipated in section 4.2, an important issue is that upon substituting the solutions of the BGT equation (40) in the expression for $|k_1, k_2\rangle$, $\langle k_1, k_2 |$, spurious divergences appear. The divergences are due to the presence of the term $(\gamma/2 - \sin(k_1) - \sin(k_2))^{-1}$. Moreover, both $N_1$ and $N_2$ in (59) and (60) diverge as well. To solve this issue, one can first exploit the invariance under reflections of $k_1, k_2$. Specifically, it is convenient to consider new quasimomenta

$$k_1 \to k_1, \quad k_2 \to -k_2. \tag{65}$$

After using (65), one obtains that the term $(\gamma/2 - \sin(k_1) + \sin(k_2))^{-1}$ is singular. This is convenient because now only $N_1$ is singular, whereas $N_2$ (cf. (59)) is regular. To proceed, one has to determine the singularities of $|k_1, k_2\rangle$, $\langle k_1, k_2 |$. The singularity structure of the terms entering in $|k_1, k_2\rangle$ (cf. (9)) is given as

$$e^{-ik_1 x_1 - ik_2 x_2} A_{12}(-,-) \simeq \delta^{1-(x_1+x_2)/(2L)}, \tag{66}$$

$$e^{-ik_1 x_1 + ik_2 x_2} A_{12}(-,+) \simeq \delta^{-(x_1-x_2)/(2L)}, \tag{67}$$

$$e^{ik_1 x_1 - ik_2 x_2} A_{12}(+,-) \simeq \delta^{1+(x_1-x_2)/(2L)}, \tag{68}$$

$$e^{ik_1 x_1 + ik_2 x_2} A_{12}(+,+) \simeq \delta^{(x_1+x_2)/(2L)}, \tag{69}$$

where we assume $x_1 \leq x_2$, and we defined $\delta$ as

$$\delta = \gamma/2 - \sin(k_1) + \sin(k_2). \tag{70}$$

Notice that $\delta = \mathcal{O}(e^{-aL})$, with $a > 0$. A similar result can be obtained for the terms with scattering amplitudes $A_{21}$ (cf. (9)). Importantly, to derive (66) (67) (68) (69), we employed the Bethe equations (18) and (19) to write the diverging contributions $e^{ik_j L}$ in terms of $\delta$. To proceed, we observe that a similar calculation can be done for the contributions appearing in $\langle k_1, k_2 |$. Now, upon taking the limit $\delta \to 0$ the singularities cancel out. Precisely, the term $N_1$ in the norm (59) diverges as $\delta^{-1}$, implying that (cf. (46)) $|k_1, k_2\rangle \langle k_1, k_2 | 0 \rangle = \mathcal{O}(\delta^{-1})$ for any $k_1, k_2$ satisfying the BGT equation (40). A straightforward calculation shows that the only possibility is that $\langle k_1, k_2 | 0 \rangle = \mathcal{O}(\delta^{-1})$ and $|k_1, k_2\rangle = \mathcal{O}(1)$. Precisely, only the term with $A_{12}(-,+)$ (cf. (9)) survives in $|k_1, k_2\rangle$. Similarly, one has to keep only the terms with $\bar{A}_{12}(+,-)$ and $\bar{A}_{21}(+,-)$ in the overlap $\langle k_1, k_2 | 0 \rangle$. After removing the singularities, the fermionic density $G_{x1,x1}$ is given as

$$G_{x_1,x_1} = \sum_{\{k_1,k_2\}} \widetilde{N}_1^{-1} e^{i(k_2-k_1)x_1} B_{12}(-,+) \bar{g}_{x,x}. \tag{71}$$

The sum in (71) is over the eigenstates of $\mathcal{L}^{(2)}$ forming the diffusive band (see section 3.5), and which are treated within the framework of the string hypothesis. In (71) we defined

$$B_{12}(-,+) = \frac{e^{ik_1}(1 - ie^{-ik_1}\gamma^-)}{e^{ik_1} - i\gamma^-}, \tag{72}$$

and $\bar{g}_{x,x}$ is the finite part of the overlap with the initial condition, and it reads as

$$\bar{g}_{x,x} = e^{i(k_1-k_2)x}\bar{B}_{12}(+,-) + e^{-i(k_1-k_2)z}\bar{B}_{21}(+,-), \tag{73}$$

where

$$\bar{B}_{12}(+,-) = \frac{1 - i\gamma^- e^{-ik_1}}{1 - i\gamma^- e^{ik_1}} \frac{\gamma/2 + \sin(k_1) - \sin(k_2)}{\gamma/2 - \sin(k_1) - \sin(k_2)}(\gamma + 2\sin(k_1) + 2\sin(k_2)), \tag{74}$$

$$\bar{B}_{21}(+,-) = -\frac{1 + i\gamma^- e^{ik_2}}{1 + i\gamma^- e^{-ik_2}}(\gamma + 2\sin(k_1) - 2\sin(k_2)). \tag{75}$$

Finally, the finite part of the normalization $\widetilde{N}_1^{-1}$ in (71) is given as

$$\widetilde{N}_1^{-1} = \frac{4i\gamma(\cos(k_1) - \cos(k_2))(\gamma + \sin(k_1) + \sin(k_2))}{(\gamma/2 + \sin(k_1) - \sin(k_2))(\gamma/2 - \sin(k_1) - \sin(k_2))}L. \tag{76}$$

Again, Eq. (71) should hold in the long-time limit, provided that $L$ is large enough to ensure the validity of the string hypothesis.

## 5 Numerical results

Here we provide numerical results supporting the Bethe Ansatz treatment of the previous sections. First, in section 5.1 we discuss the numerical solution of the Bethe equations (18) and (19). In section 5.2 we focus on the Bethe-Gaudin-Takahashi equation (37) and (40). In section 5.3 we discuss exact diagonalization data for the eigenvalues of $\mathcal{L}^{(2)}$. In section 5.4 we compare the spectrum of $\mathcal{L}^{(2)}$ obtained from exact diagonalization with the Bethe Ansatz results. In section 5.5 we investigate the finite-size scaling of the Liouvillian gap. In section 5.6 we address the dynamics of the *full* correlator $G_{x_1,x_2}$ (cf. (6)). In section 5.7 we focus on the profile of the fermion density. Finally, in section 5.8 we discuss the diffusive scaling of the fermion density and its violation due to the boundary losses.

### 5.1 Numerical solution of the Bethe equations

The numerical solution of the Bethe equations is in general a challenging task. Indeed, Eq. (18) and (19) have $L^2$ solutions in the complex plane. Moreover, $k_1 = 0$ and $k_2 = 0$, as well as $k_1 = \pi$ and $k_2 = \pi$ are always solutions, although they have to be excluded because they correspond to vanishing eigenvectors. Similarly, the solutions with $k_1 = k_2$ have to be excluded. Pairs of solutions $(k_1, k_2)$ that are related by a shift by $\pm\pi$ have to be counted only once. Crucially, since all the solutions of (18) and (19) are complex, multiple-precision arithmetic is necessary to evaluate the exponentials in the left-hand side of the equations.

To illustrate the structure of the solutions of the Bethe equations, it is useful to focus on chains with small $L$. In Table 1 we show the full set of solutions of (18) and (19) for $L = 2$ and $L = 3$. We only a consider fermionic chain with open boundary conditions and $\gamma = \gamma^- = 1/10$. The second column of the table shows the eigenvalues of $\sigma$ (cf. (9)). The states with $\sigma = -1$ are not affected by dephasing, as discussed in section 3.3. The number of solutions with

Table 1: Full set of solutions of the Bethe equations (cf. (18)(19)) for with $L = 2$ (first four rows) and $L = 3$ (last nine rows) sites, and with $\gamma = \gamma^- = 1/10$. We show the two quasimomenta $k_1, k_2$ and the associated energy $\varepsilon$. The last solution for $L = 2$ and the last three solutions for $L = 3$ correspond to $\sigma = -1$ in (9), i.e., they are the same as for $\gamma = 0$. Notice that by using the symmetries of the Bethe equations, we fix $0 < \text{Re}(k_1) < \pi$ and $0 < \text{Re}(k_2) < \pi$. Given a set of solutions $(k_1, k_2)$, one has that $(k_2^*, k_1^*)$ is also a solution. Notice that for $\sigma = -1$ only $L(L-1)/2$ solutions are allowed.

| $L$ | $\sigma$ | $k_1$ | $k_2$ | $\varepsilon$ |
|---|---|---|---|---|
| 2 | + | $2.1919997294 - 0.1312558459i$ | $1.1457120114 - 0.0351640458i$ | $-0.25 - 1.9993749023i$ |
| 2 | + | $1.1457120114 + 0.0351640458i$ | $2.1919997294 + 0.1312558459i$ | $-0.25 + 1.9993749023i$ |
| 2 | + | $0.7866450552 - 0.0353040029i$ | $0.7866450552 + 0.0353040029i$ | $-0.2$ |
| 2 | − | $2.0934365739 - 0.0576711461i$ | $2.0934365739 + 0.0576711461i$ | $-0.3$ |
| 3 | + | $1.28745845407 - 0.04147721049i$ | $1.28745845407 + 0.04147721049i$ | $-0.2593393435$ |
| 3 | + | $2.67275603750 + 0.00860942154i$ | $2.67275603750 - 0.00860942154i$ | $-0.1155608174$ |
| 3 | + | $2.28508055892 + 0.03633954446i$ | $0.71654617450 + 0.08931362935i$ | $-0.162549919 - 2.8251952277i$ |
| 3 | + | $0.71654617450 - 0.08931362935i$ | $2.28508055892 - 0.03633954446i$ | $-0.162549919 + 2.8251952277i$ |
| 3 | + | $2.36338604213 - 0.06913029032i$ | $1.57756565342 + 0.0014316741i$ | $-0.2 - 1.4142135623i$ |
| 3 | + | $1.5775656534 - 0.00143167416i$ | $2.36338604213 + 0.0691302903i$ | $-0.2 + 1.4142135623i$ |
| 3 | − | $0.7866450552 - 0.0353040029i$ | $0.78664505521 + 0.03530400298i$ | $-0.2$ |
| 3 | − | $1.5707963267 - 0.0499791900i$ | $2.35494759837 + 0.03530400298i$ | $-0.25 + 1.4133294025i$ |
| 3 | − | $2.3549475983 - 0.0353040029i$ | $1.57079632679 + 0.04997919006i$ | $-0.25 - 1.4133294025i$ |

$\sigma = -1$ is $L(L-1)/2$. The third and fourth column show the solutions $(k_1, k_2)$ of the Bethe equations. Importantly, the invariance of (18) and (19) under the change of sign of $k_j$ and under shifts by $\pi$ can be used to fix $0 < \text{Re}(k_j) < \pi$. Notice also that the Bethe equations are not invariant under exchange $k_1 \leftrightarrow k_2$. However, given a solution $(k_1, k_2)$, then $(k_2^*, k_1^*)$, with the star denoting complex conjugation, is also a solution. As it is clear from (1), some of the solutions are formed by pairs of complex conjugated momenta $(k_1, k_1^*)$. These correspond to the real eigenvalues $\varepsilon$ of $\mathcal{L}^{(2)}$. The last column in Table 1 is the energy $\varepsilon$ as obtained by using (11). We checked that the eigenvalues coincide with the exact diagonalization results to machine precision.

It is interesting to investigate how the solutions of the Bethe equations change as a function of dissipation. In Fig. 4 we show the solutions of the Bethe equations for $L = 2$ at fixed $\gamma^- = 1/10$ as a function of $\gamma$. We only consider the three solutions with $\sigma = 1$ (cf. (9) and Table 1). Panel $(a)$ and $(b)$ show $\text{Re}(k_j)$ and $\text{Im}(k_j)$ as a function of $\gamma$. We consider the interval $\gamma \in [1/10, 6]$. We denote the different solutions as $(k_1^{(p)}, k_2^{(p)})$, with $p \in [1, 3]$. Now, the solution at the bottom in Fig. 4 (a), i.e., with $p = 3$, corresponds to $k_2 = k_1^*$, i.e., to real energy $\varepsilon$. The remaining two solutions are such that $k_1^{(1)} = (k_2^{(2)})^*$ and $k_1^{(2)} = (k_1^{(1)})^*$. Interestingly, the behavior of the solutions as a function of $\gamma$ is not "smooth". Precisely, at $\gamma = 4$ the solutions with $p = 1$ and $p = 2$ "collide", whereas the one with $p = 3$ remains isolated. At $\gamma > 4$ the solutions with $p = 1, 2$ get reorganized. Precisely, they emerge as new pairs of solutions $(\tilde{k}_1^{(p)}, \tilde{k}_2^{(p)})$, with $p = 1, 2$ for $\gamma > 4$. Notice that for $\gamma > 4$ all the three solutions are formed by complex conjugated momenta, i.e., $\tilde{k}_1^{(p)} = (\tilde{k}_2^{(p)})^*$ for any $p$.

## 5.2 Numerical solution of the Bethe-Gaudin-Takahashi (BGT) equation

The numerical results of the previous section showed that extracting the full set of solutions of the Bethe equations can be a challenging task, as expected. Here we focus on the solutions of the Bethe equations forming the diffusive band (see Fig. 2 and section 3.5). These solutions dominate the long-time behavior of physical observables, such as the fermion correlator $G_{x_1, x_2}$. In the limit of large $L$, one can use the string hypothesis [54]. Thus, the solutions of the Bethe

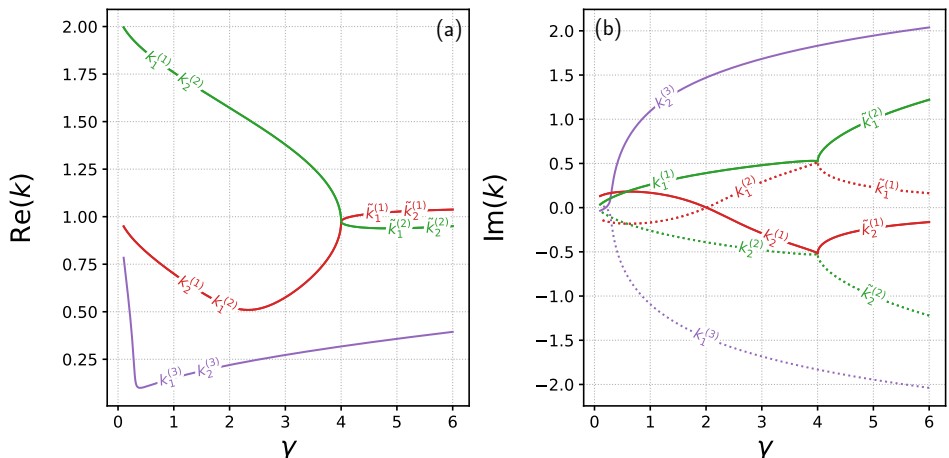

Figure 4: Solutions $(k_1, k_2)$ of the Bethe equations (18) and (19) for $L = 2$ and fixed $\gamma^- = 1/10$. We only consider solutions with $\sigma = 1$ (cf. (9)). We plot $(k_1, k_2)$ as functions of the dephasing rate $\gamma$. Panel $(a)$ shows the real parts of $k_1$ and $k_2$, the imaginary parts being reported in panel $(b)$. Starting from $\gamma = 0$ one has the three pairs $(k_1^{(j)}, k_2^{(j)})$ with $j = 1, 2, 3$. The solutions with $j = 1, 2$ are related by $(k_1^{(1)}, k_2^{(1)}) = (k_2^{(2)}, k_1^{(2)})^*$, with the star denoting complex conjugation. The solution with $j = 3$ is such that $k_1^{(3)} = (k_2^{(3)})^*$. At $\gamma = 4$ the solutions with $j = 1, 2$ "collide". The solutions at $\gamma > 4$ are denoted with a tilde. At $\gamma > 4$ all the solutions are formed by complex conjugate pairs, i.e., $\tilde{k}_1^{(j)} = (\tilde{k}_2^{(j)})^*$ for any $j$.

equations forming the diffusive band are well approximated by the solutions of the Bethe-Gaudin-Takahashi equation (37). Solving (37) is a much simpler task because Eq. (37) is only function of $k_{\mathrm{im}}$, which is real. Importantly, by using the BGT equation in logarithmic form (40) and by varying the quantum numbers $I_j$, one can target the different momentum pairs $(k_1, k_2)$ forming the diffusive band.

Here we focus on the numerical solution of the BGT equation (37). In Fig. 5 we plot the real and imaginary parts (curves with different colors) of (37) for a chain with $L = 10$, $\gamma = 5$ and $\gamma^- = 0$. On the $x$-axis $k_{\mathrm{im}}$ is the imaginary part of $k_1 = k_2^*$. The real part is obtained from (35). The simultaneous crossing (full circles) of the two curves with the horizontal axis marks the solutions of the BGT equation. Notice that there are $L - 1$ solutions. The missing solution is that with $\varepsilon = 0$. This is present only for $\gamma^- = 0$ and it corresponds to diverging $k_1, k_2$. Let us now investigate the effect of the losses. In Fig. 6 we show the numerical solutions of (37) for $L = 10$, $\gamma = 5$ and $\gamma^- = 1$. Now, there are $L$ solutions. The solution with $\varepsilon = 0$ is not present. Interestingly, the leftmost circle in Fig. 6 corresponds to two almost degenerate solutions of (37). These are the boundary-related eigenvalues of $\mathcal{L}^{(2)}$ discussed in section 3.5. These boundary states are present only for $\gamma^- > \exp(-\mathrm{arccosh}(\gamma/4))$. Upon lowering $\gamma^-$ they merge with the diffusive band (cf. 2). The inset of Fig. 6 shows a zoom of the real and imaginary parts of the BGT equation (37) around the two degenerate solutions (leftmost circle in the main Figure). For $L = 10$ the difference between the two solutions is $\mathcal{O}(10^{-4})$.

## 5.3 Spectrum of the Liouvillian $\mathcal{L}^{(2)}$: Overview

Here we illustrate the general structure of the spectrum of the Liouvillian $\mathcal{L}^{(2)}$ for a fermionic chain with open boundary conditions with $\gamma^- = 0$ as a function of $\gamma$. In Fig. 7 we report exact diagonalization results for a chain with $L = 20$ and no boundary losses, i.e., with $\gamma^- = 0$. First, for $\gamma = \gamma^- = 0$ the $L^2$ eigenvalues $\varepsilon$ form a straight line parallel to the imaginary axis (not

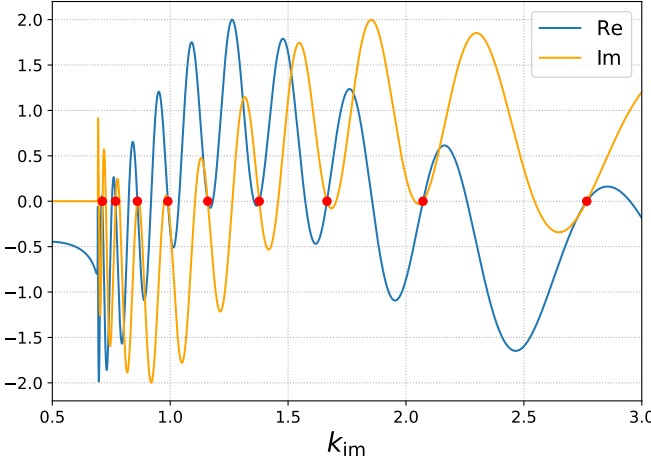

Figure 5: Numerical solution of the BGT equation (37) for a fermionic chain with $L = 10$ and $\gamma = 5$ and $\gamma^- = 0$. The two curves show the real and imaginary parts of (37), plotted versus the imaginary part $k_{\text{im}}$ of the quasimomenta $k_1, k_2$. The circles are the solutions of the BGT equations. Notice that there are $L - 1$ solutions. The missing solution gives $\varepsilon = 0$, and it corresponds to $k_{\text{im}} \to \infty$.

shown in the Figure). Indeed, for $\gamma = 0$, $k_1, k_2$ are solutions of (24) and (25) with $\gamma^- = 0$, and are real. As a consequence the eigenvalues $\varepsilon$ (cf. (11)) have the same real part $-\gamma$. As discussed in section 3.3, at finite $\gamma$ there are $L(L-1)/2$ momentum pairs $(k_1, k_2)$ that remain the same as for $\gamma = 0$. This implies that the eigenvalues $\varepsilon$ are the same as for $\gamma = 0$, apart from a trivial shift by $-\gamma$ (cf. (11)). These eigenvalues correspond to the vertical straight lines with $\text{Re}(\varepsilon) = -\gamma$ in the different panels. Near this vertical lines there are $\sim L(L-1)/2$ eigenvalues that depend in a nontrivial way on $\gamma$. These eigenvalues correspond to complex solutions of the Bethe equations (18) and (19) with vanishing imaginary parts in the limit $L \to \infty$. For large $L$ these eigenvalues can be understood perturbatively in $1/L$, as it was discussed in section 3.4, at least for large enough $\gamma$. Finally, upon increasing $\gamma$ a band of real eigenvalues appears. This band contains the eigenvalues that are responsible for the diffusive spreading of particles at long times. A similar band appears in the tight-binding chain with periodic boundary conditions [26], and in the periodic tight-binding chain subject to incoherent hopping [28]. As $\gamma$ increases, the separation in energy between the diffusive band and the remaining part of the spectrum increases. Similar separation in different connected components of the Liouvillian spectrum has been observed in random Liouvillians [57]. As discussed in section 3.5, at $\gamma > 4$ the diffusive band contains at least $L - 2$ states, whereas the number of states diminishes upon lowering $\gamma$. These eigenvalues in the diffusive band correspond to string solutions of the Bethe equations (18) and (19), and can be effectively treated within the framework of the string hypothesis (see section 3.5).

## 5.4 Spectrum of $\mathcal{L}^{(2)}$: Exact diagonalization versus Bethe Ansatz

Let us now compare the Bethe Ansatz results for the eigenvalues $\varepsilon$ of the Liouvillian $\mathcal{L}^{(2)}$ and exact diagonalization (ED) data. In Fig. 8 we show ED data for $L = 20$, $\gamma^- = 0$ and $\gamma = 3$ and $\gamma = 5$ (left and right panel, respectively). In both panels there is a vertical band containing $L(L-1)/2$ eigenvalues. These are the same, except for a trivial shift by $-\gamma$, as for the open fermionic chain with $\gamma = 0$. At small $\text{Re}(\varepsilon)$ the diffusive band of real eigenvalues is visible. The complex eigenvalues $\varepsilon$ between the diffusive band and the vertical band correspond to complex solutions of the Bethe equations with vanishing imaginary parts in the limit $L \to \infty$.

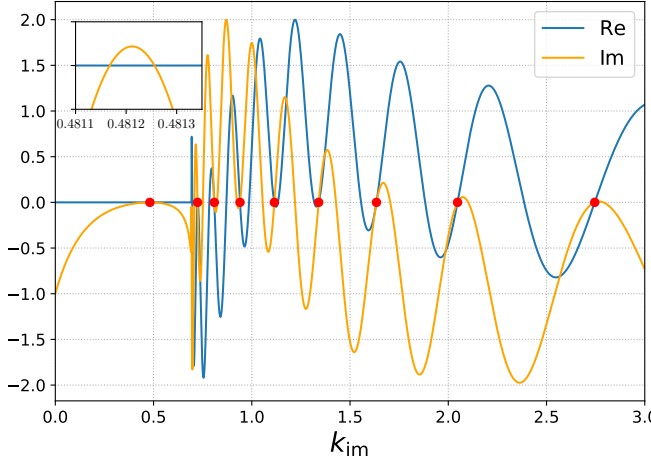

Figure 6: Same as in Fig. 5 for $\gamma^- = 1$. Now there are $L - 2$ solutions within the diffusive band. The solution with $\varepsilon = 0$ is not present for nonzero $\gamma^-$. The two almost degenerate solutions at $k_{im} \approx 0.5$ (see inset in the Figure) correspond to the boundary-related eigenvalues of the Liouvillian (star symbol in Fig. 2 (b)).

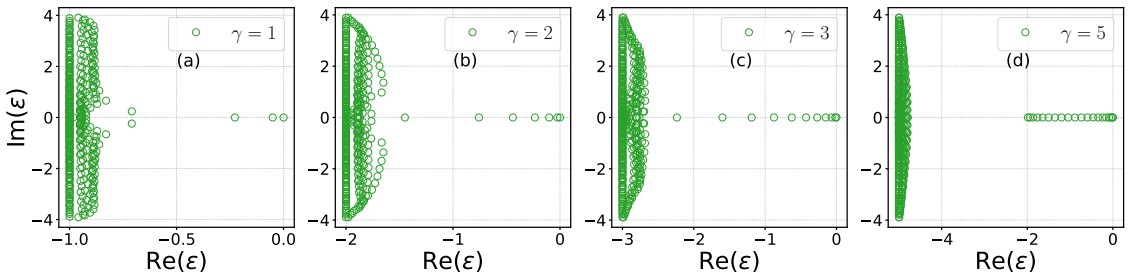

Figure 7: Spectrum of the Liouvillian $\mathcal{L}^{(2)}$ for a fermionic chain with bulk dephasing and no boundary losses, i.e., $\gamma^- = 0$. The symbols are exact diagonalization results for $L = 20$. We plot $\text{Im}(\varepsilon)$ versus $\text{Re}(\varepsilon)$. The different panels show different values of $\gamma$. As one increases $\gamma$ (from left to right in the Figure), the diffusive band with real eigenvalues gets populated and well separated from the rest of the spectrum.

As discussed in section 3.4 these eigenvalues of $\mathcal{L}^{(2)}$ can be understood perturbatively in $1/L$. The full circles in the Figure are obtained from the large $L$ expansions (30) and (31). Fig. 8 shows that the large $L$ expansion works well at $\gamma = 5$, i. e., when the diffusive band is well separated from the rest of the spectrum. However, when the diffusive band overlaps with the other regions of the spectrum, the agreement between the large $L$ expansions and the ED data is not perfect.

For $\gamma = 5$ in Fig. 8 we report with the crosses the Bethe Ansatz results for the eigenvalues of the diffusive band. As it was stressed in section 5.1, extracting the full set of solutions of the Bethe equations is a challenging task. A convenient strategy for the diffusive band is to first solve the BGT equation (37), and then use the solutions as initial guess to solve (18) and (19). The agreement between the Bethe Ansatz results and the ED data is perfect. A similar agreement is found for $\gamma = 3$, although we do not report the results in the Figure. Finally, let us discuss the effect of $\gamma^-$. In Fig. 9 we show ED results for $L = 20$, $\gamma = 5$ and $\gamma^- = 1$. Now, the vertical band at small eigenvalues $\varepsilon \approx -\gamma$ is deformed. Still, there are $L(L-1)/2$ eigenvalues that are the same as in the case with $\gamma = 0$, apart from the trivial shift by $-\gamma$. Again, a diffusive band is present at large $\text{Re}(\varepsilon) \approx 0$. The band contains $L$ energies.

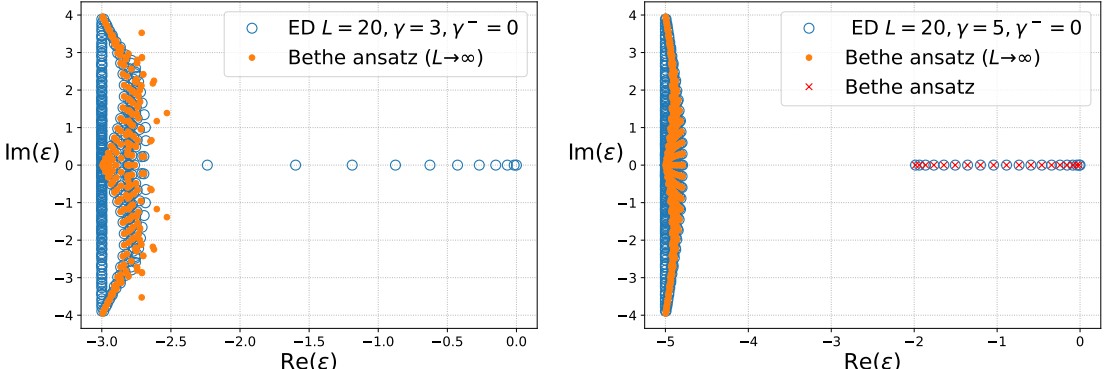

Figure 8: Spectrum of the Liouvillian $\mathcal{L}^{(2)}$ for the open fermionic chain with $L = 20$ and $\gamma^- = 0$. The left (right) panel shows exact diagonalization results (circles) for $\gamma = 3$ ($\gamma = 5$). In both panels a vertical band of $L(L-1)/2$ eigenvalues is present at $\varepsilon = -\gamma$. These eigenvalues are the same as for $\gamma^- = 0$, apart from a trivial shift by $-\gamma$. Near the vertical band a connected region with $\sim L^2/2$ eigenvalues is present. This correspond to the complex solutions of the Bethe equations with vanishing imaginary parts (see section 3.4). Finally, a diffusive band of real eigenvalues is also visible in both panels. For $\gamma = 5$ (right panel) the diffusive band is well separated from the rest of the spectrum, and it contains $L$ states. The large $L$ expansion (30) and (31) is reported in both panels with the full circles. The expansion describes well the eigenvalues near the vertical band. The crosses in the right panel are Bethe Ansatz results obtained by solving numerically (32) and (33) using the solutions of the BGT equation (40) as initial guess.

The full diamonds are the results obtained using the string hypothesis, i.e., by solving the BGT equation (37). The agreement with the ED data is perfect, even at finite $\gamma^-$ and despite the fact that the BGT equation (37) is valid only in the limit $L \to \infty$. Finally, the real energy at $\varepsilon \approx -2.7$ corresponds to the almost degenerate doublet of boundary-related eigenvalues of the Liouvillian (see section 3.5), which appear for $\gamma^- > \exp(-\text{arccosh}(\gamma/4))$.

## 5.5 Finite-size scaling of the Liouvillian gap

Let us now discuss the finite-size scaling of the gap of the Liouvillian $\mathcal{L}^{(2)}$. The Liouvillian gap $\Delta\mathcal{L}^{(2)}$ is the eigenvalue of $\mathcal{L}^{(2)}$ with the largest *nonzero* real part, i.e.,

$$\Delta\mathcal{L}^{(2)} := -\max_j \text{Re}(\varepsilon_j), \quad \text{with } \text{Re}(\varepsilon_j) \neq 0. \tag{77}$$

As it is clear from Fig. 8 and Fig. 9, the gap coincides with the largest nonzero energy in the diffusive band. For $\gamma^- = 0$ and in the large $L$ limit the gap is obtained by solving the BGT equation (40) with $I_j = 1$. For $\gamma^- > 0$ one has to fix $I_j = 0$ in (40). In Fig. 10 we plot the Liouvillian gap as a function of $L$. We show results for $\gamma = 3, 5$ and $\gamma^- = 0, 1$. The symbols are the exact numerical data obtained from the BGT equation (40). The dashed-dotted lines are the results (44) in the large $L$ limit. The leading behavior as $\propto 1/L^2$ is visible. Notice that although Eq. (44) was derived for the case with nonzero $\gamma^-$ (see section 3.5), it works also for $\gamma^- = 0$. In the inset in Fig. 10 we focus on subleading terms, subtracting from $\Delta\mathcal{L}^{(2)}$ the leading $1/L^2$ behavior (cf. (44)). Precisely, we plot $\Delta\mathcal{L}^{(2)} + 2\pi^2/(\gamma L^2)$ versus $L$. We only consider the case with $\gamma = 5$ and $\gamma^- = 1$. The continuous line is the second term in (44), which perfectly matches the data.

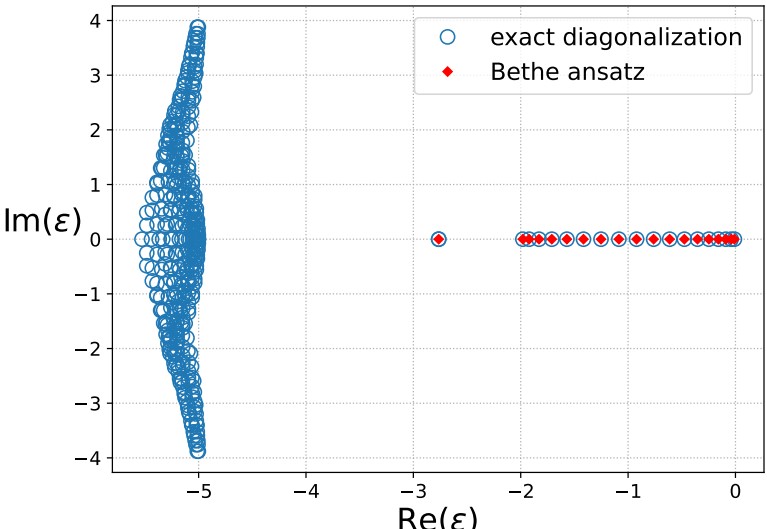

Figure 9: Same as in Fig. 8 for $L = 20$, and $\gamma = 5$ and $\gamma^- = 1$. The diffusive band with $\varepsilon > \sqrt{\gamma^2 - 16} - \gamma$ contains $L - 2$ solutions. An isolated doublet of quasi-degenerate eigenvalues is present at $\varepsilon \approx -2.7$, and it corresponds to the boundary-related eigenvalues in Fig. 2 (b) (see full diamond symbol in the Figure). The empty circles in the Figure are exact diagonalization (ED) results. The diamonds are the results obtained by solving the BGT equation (37) numerically.

## 5.6 Dynamics of the fermionic correlator: Bethe Ansatz versus exact diagonalization

Having compared Bethe Ansatz versus exact diagonalization results for the spectrum of the Liouvillian, we now focus on the time-dependent fermionic correlator $G_{x_1,x_2}(t)$. Here we compare ED data versus Bethe Ansatz results for the *full* correlator $G_{x_1,x_2}$. The circles in Fig. 11 are ED data for a chain with $L = 10$ sites, $\gamma = 5$, and $\gamma^- = 1$. The results are at fixed time $t = 20$. The left and right panels in the Figure show $\mathrm{Re}(G_{x_1,x_2})$ and $\mathrm{Im}(G_{x_1,x_2})$, respectively. Notice that on the $y$-axis we employ a logarithmic scale. The numbers on the real axis label the different matrix elements of $G_{x_1,x_2}$. The full diamonds are Bethe Ansatz results. Precisely, here we obtain the full-time dynamics of $G_{x_1,x_2}$ by using the expansion (46). However, since finding all the solutions of the Bethe equations (18) and (19) is a daunting task, we truncate (46) restricting the sum over the eigenvalues in the diffusive band, which are expected to dominate the long-time behavior of the correlator. We first numerically solve the BGT equation (40), using the solutions as initial guess for the exact Bethe equations (18) and (19). As it is clear from Fig. 11, the agreement between the Bethe Ansatz and the exact diagonalization data is quite satisfactory. Deviations are present for the smaller matrix elements, and can be attributed to the complex eigenvalues $\varepsilon$ of the Liouvillian, which we are neglecting.

## 5.7 Dynamics of the density profile

Here we address the long-time limit of the fermionic density profile, i.e., the diagonal correlators $G_{x,x}$. This is investigated in Fig. 12. In the left and right panels we show the dynamics of $G_{L/2,L/2}$ and $G_{1,1}$, respectively. We focus on a chain with $L = 10$ sites. The data are for $\gamma = 5$ and $\gamma^- = 1$. The circles in the figures are exact diagonalization results. The dashed line is the Bethe Ansatz result obtained by using (46), where we restrict the sum to the eigenvalues in the diffusive band (see Fig. 9). Importantly, we use the solutions of the Bethe equations (18)

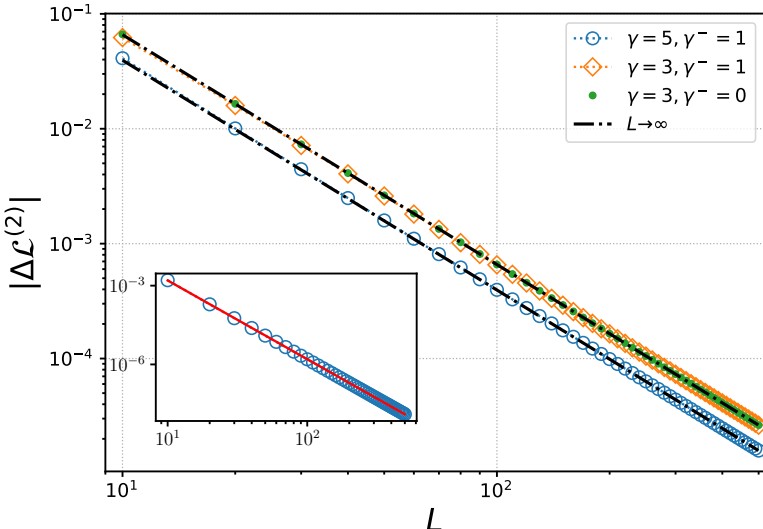

Figure 10: Finite-size scaling of the Liouvillian gap defined in (77). We show results for several values of the dephasing rate $\gamma$ and boundary loss rate $\gamma^-$ (different symbols in the Figure). We consider chains with $L \lesssim 500$. The dashed-dotted line is the Bethe Ansatz prediction (44) in the limit $L \to \infty$. In the inset we focus on the subleading contributions to the gap for the case with $\gamma = 5$ and $\gamma^- = 1$. The symbols are $\Delta\mathcal{L}^{(2)} + 2\pi^2/(\gamma L^2)$ plotted versus $L$. The continuous line is the analytic prediction (cf. second term in (44)).

and (19). Let us first focus on $G_{L/2,L/2}$. At $t = 0$ one has that $G_{L/2,L2} = 1$, while $G_{L/2,L/2}$ vanishes for $t \to \infty$. The agreement between the ED data and the Bethe Ansatz is remarkable. Deviations are visible at short times. This is expected because of the truncation in (46). At short times the contribution of the complex eigenvalues of the Liouvillian cannot be neglected. The continuous line in Fig. 12 is obtained by considering in (46) only the contribution of the Liouvillian gap, i.e., the energy $\varepsilon$ with the largest nonzero real part. The scenario is slightly different for $G_{1,1}$ (right panel in Fig. 12). Precisely, $G_{1,1}$ is zero at $t = 0$, it increases at later times as the particle initially at $x = L/2$ spreads towards the edges. At long times the dynamics is dominated by the boundary loss, and $G_{1,1}$ vanishes. As for $G_{L/2,L/2}$ the agreement between the ED data and the Bethe Ansatz obtained by using the diffusive band states (dashed line) is quite satisfactory, although at intermediate times is only qualitatively accurate. On the other hand, the approximation obtained by restricting the sum in (46) to the Liouvillian gap works only at long times.

The profile of the fermionic density $G_{x,x}$ at fixed time and as a function of $x$ is reported in Fig. 13. We show results for a chain with $L = 10$ and time $t = 20$ (empty circles) and $t = 40$ (empty squares). The full diamonds are Bethe Ansatz results obtained from (46) restricting the sum over the states in the diffusive band but using the exact solutions of the Bethe equations (18) and (19). The agreement between the Bethe Ansatz and the exact diagonalization data is excellent for any $x$. The hexagons symbols show the Bethe Ansatz results obtained employing the string hypothesis, i.e., by using the results of section 4.3. The agreement with the ED data is satisfactory, although some deviations are visible.

## 5.8 Diffusive scaling

In the long-time limit the density profile $G_{x,x}$ should exhibit diffusive scaling, at least if time is short enough that we can neglect the effect of the boundary losses. This diffusive behavior

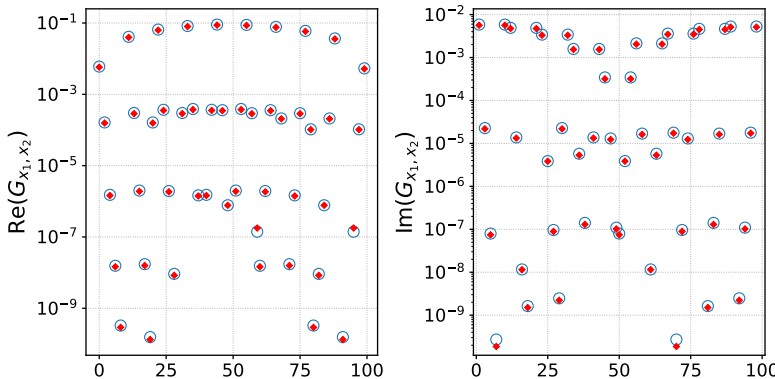

Figure 11: Fermionic correlator $G_{x_1,x_2}$ as a function of time $t$. The left and right panels show the real and imaginary parts of the correlator, respectively. The initial condition is $G_{x_1,x_2} = \delta_{x_1,L/2}\delta_{x_2,L/2}$. The empty circles are exact diagonalization (ED) results for the matrix elements of $G_{x_1,x_2}$ at $t = 20$. Results are for a chain with $L = 10$, $\gamma = 5$ and $\gamma^- = 1$. The full diamonds are the Bethe Ansatz results. These are obtained by solving the Bethe equations (18) and (19) and using (46), where the sum is restricted to the eigenvalues of the diffusive band.

is observed in the periodic chain [26]. This is investigated in Fig. 14 focusing on the late-time dynamics of $G_{x,x}$ starting from the initial condition with a fermion localized at the center of the chain. In Fig. 14 (a) we show $G_{x,x}$ as a function of $x - L/2$. The symbols are data at different times, and are obtained by using the results of section 4.3. The data are for a system with $L = 100$ sites. As it is clear from the Figure, the fermion density spreads diffusively as time increases, reaching the boundary of the chain at late times. Precisely, in the diffusive regime $G_{x,x}$ is given by

$$G_{x,x} = \frac{1}{\sqrt{4\pi D t}} \exp\left[-\frac{(x-L/2)^2}{4Dt}\right], \quad D := \frac{2}{\gamma}, \tag{78}$$

with $D$ the diffusion constant, which was derived in Ref. [26]. The diffusive scaling is investigated in Fig. 14 (b), plotting $t^{1/2}G_{x,x}$ versus $(x-L/2)/t^{1/2}$. Up to $t = 80$ all the data collapse on the same curve, which is in perfect agreement with (78). At longer times the effect of the boundary loss is non negligible and the diffusive scaling breaks down. At times $t \gg L^2$ the fermion density is vanishing at the edges of the chain, and the height of the fermionic lump that is left around the center of the chain diminishes with time.

## 6 Conclusions

We derived the Bethe Ansatz for the spectrum of the Liouvillian $\mathcal{L}^{(2)}$, which determines the dynamics of the fermionic correlator $G_{x_1,x_2}$ in the fermionic tight-binding chain in the presence of bulk dephasing and boundary losses. For large enough dephasing, the spectrum of the Liouvillian comprises three different parts. Precisely, there are $L(L-1)/2$ complex eigenvalues that are trivially related to those of the tight-binding chain with boundary losses and no bulk dephasing. For this reason we dub them dephasing-indepedent eigenvalues. Furthermore, there are $\sim L(L-1)/2$ complex eigenvalues that are perturbatively related to the dephasing-independent ones in the large chain limit. Finally, there is band of $\sim L$ real eigenvalues. Since the band contains the eigenvalues with the largest real parts, it dominates the long-time behavior of the correlator, and determines the diffusive scaling at intermediate times.

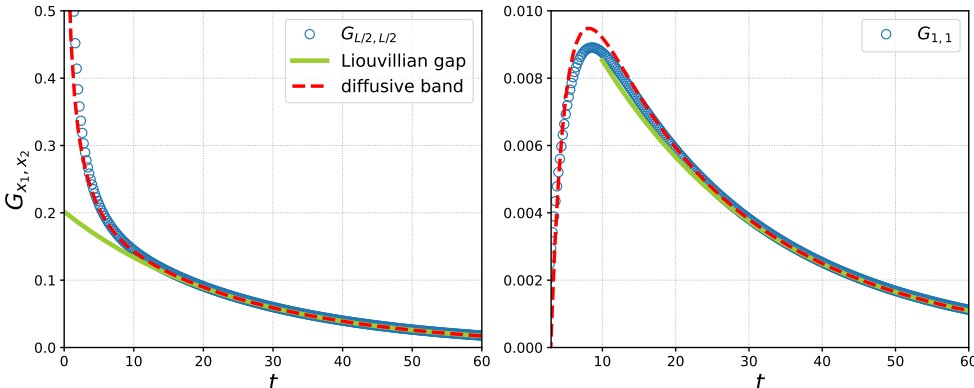

Figure 12: Evolution of the fermionic correlator $G_{x_1,x_2}$ as a function of time $t$. We show results for a chain with $L = 10$ sites, $\gamma = 5$ and $\gamma^- = 1$. The correlator at $t = 0$ is $G_{x_1,x_2} = \delta_{x_1,L/2}\delta_{x_2,L/2}$, i.e., a fermion localized at the center of the chain. The left and right panels show $G_{L/2,L/2}$ and $G_{1,1}$, respectively. The circles are exact diagonalization data. The dashed line is the Bethe Ansatz result obtained by solving numerically the Bethe equations (18) and (19), and using (9). In evolving $G_{x_1,x_2}$ we only considered the eigenvalues in the diffusive band (see Fig. 2), which explains the deviations from the ED data. The continuous line is the result considering only the energy with the largest nonzero real part.

For this reason we dub it diffusive band. Interestingly, for large enough loss rate boundary-related modes of the Liouvillian appear. Both the diffusive band and the boundary modes can be characterized by using the framework of the string hypothesis. Crucially, the Bethe Ansatz allowed us to obtain the time-dependent fermionic correlator $G_{x_1,x_2}$. In particular, we provided analytic formulas for the long-time behavior of the fermionic density $G_{x,x}$.

Let us now illustrate some possible directions for future work. First, we showed that despite the Liouvillian $\mathcal{L}^{(2)}$ being diagonalized by Bethe Ansatz, the full Liouvillian is mapped to the Hamiltonian of the open Hubbard chain with boundary magnetic fields and boundary pair production, which is not integrable. It would be interesting to investigate whether the Liouvillian $\mathcal{L}^{(4)}$ that describes the dynamics of the fermionic four point function can be diagonalized by the Bethe Ansatz. Furthermore, we showed that at long times the diffusive scaling of the fermionic density is broken, due to the boundary losses. It would be interesting to further investigate this regime to understand whether any universal scaling behavior can be extracted. An interesting direction would be to employ the Bethe Ansatz framework to characterize the interplay between dissipation and criticality [2]. This would require to extend the results of section 4 to non-diagonal initial conditions. Recently, it has been shown that several one-dimensional out-of-equilibrium systems exhibit the so-called quantum Mpemba effect [58]. It would be interesting to investigate how the Mpemba effect is affected by dissipation. While this issue has been addressed numerically (see for instance Ref. [59]), the Bethe Ansatz would allow to clarify the scenario analytically.

The full Liouvillian describing dephasing dissipation is not quadratic in the fermion operators. This implies that entanglement-related quantities are not fully determined by the two-point fermionic correlation function, in contrast with quadratic models [60]. Still, it was observed in Ref. [38] that the "entanglement entropies" defined from the fermionic correlator exhibit scaling behavior in the weak-dissipation hydrodynamic limit. Our results could allow to clarify the origin of this scaling. Finally, it would be important to understand whether the tight-binding chain with localized dephasing can be solved by Bethe Ansatz, paving the way to characterize analytically entanglement scaling [43, 44, 61, 62].

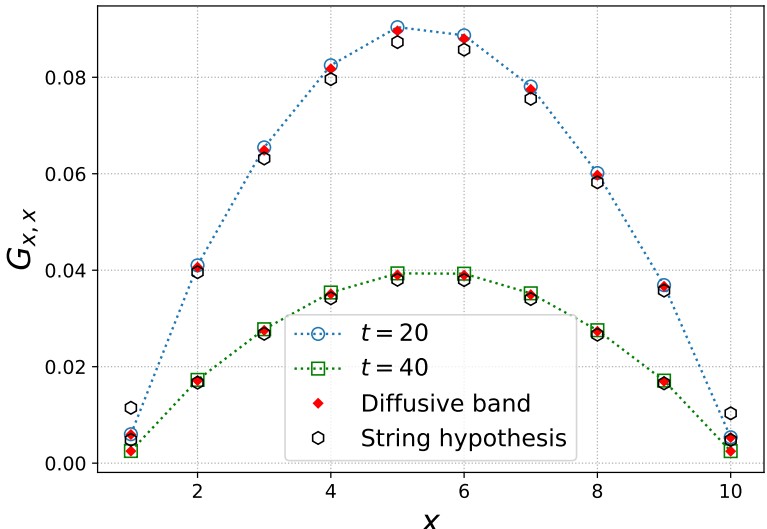

Figure 13: Density profile $G_{x,x}$ plotted as a function of the position $x$ is the chain. The results are for a chain with $L = 10$, dissipation rate $\gamma = 5$ and boundary loss rate $\gamma^- = 1$. The empty circles are exact diagonalization results for $t = 20$ and $t = 40$. At $t = 0$ the correlator is $G_{x_1,x_2} = \delta_{x_1,L/2}\delta_{x_2,L/2}$. The empty hexagons are Bethe Ansatz results in the limit $L \to \infty$. These are obtained by solving the BGT equation (40) and using (71). The full diamonds are the same Bethe Ansatz results as in Fig. 11.

## Acknowledgments

I would like to thank Yuan Miao for useful discussions. I would also like to thank the Kavli Institute for the Physics and Mathematics of the Universe (Kavli IPMU), where part of this work was completed, for the kind hospitality.

## A  Derivation of the full Liouvillian

Here we derive the full Liouvillian $\mathcal{L}$ governing the evolution of the density matrix $\rho$ (cf. (2)). We employ the formalism of the third quantization [7]. This will allow us to map the Liouvillian to a one-dimensional system of spinful fermions described by a Hubbard-like Hamiltonian. In section A.1 we compare the result with the Hamiltonian of the one-dimensional Hubbard model with boundary magnetic fields.

The action of the Liouvillian on a generic density matrix can be understood by using the formalism of Ref. [7]. A generic density matrix can be decomposed as a superposition of strings of Majorana operators $\Gamma_{\underline{\nu}}$ defined as

$$\Gamma_{\underline{\nu}} := a_1^{\nu_1} a_2^{\nu_2} \cdots a_{2L}^{\nu_{2L}}, \tag{A.1}$$

where $a_j$ are Majorana fermionic operators with standard anticommutation relations $\{a_j, a_k\} = 2\delta_{jk}$, and $\nu_j = 0, 1$ occupation numbers. The string of operators in (A.1) is *ordered*. The relationship between Majorana fermions $a_j$ and Dirac fermions $c_j$ is given as

$$a_{2j-1} = c_j + c_j^\dagger, \quad a_{2j} = i(c_j - c_j^\dagger). \tag{A.2}$$

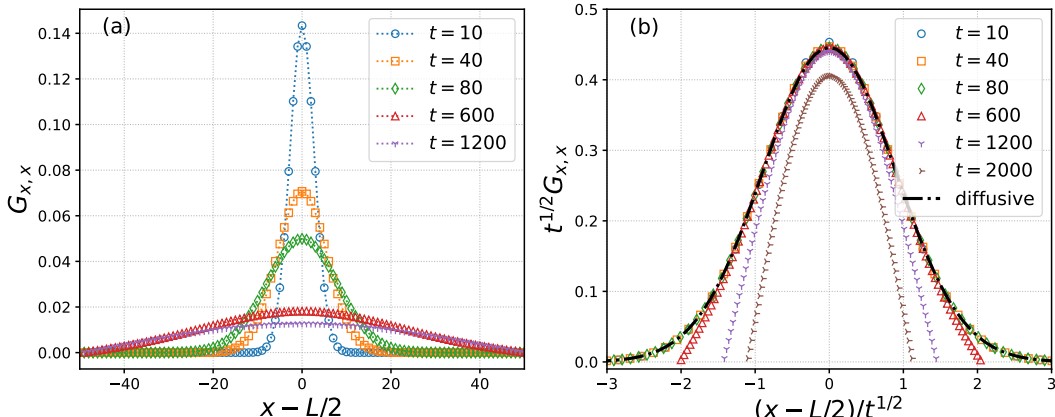

Figure 14: Dynamics of the density profile $G_{x,x}$ for a fermionic chain with $L = 100$, dephasing rate $\gamma = 5$, and boundary loss rate $\gamma^- = 1$. Panel $(a)$ shows the time-dependent correlator $G_{x,x}$ as a function of $x - L/2$. At $t = 0$, $G_{x_1,x_2} = \delta_{x_1,L/2}\delta_{x_2,L/2}$. Panel $(b)$ shows the rescaled density $t^{1/2}G_{x,x}$ as a function of $(x - L/2)/t^{1/2}$. For $t < 600$ the dynamics exhibits a clear diffusive scaling. At later times the boundary losses affect the dynamics, and the diffusive scaling breaks down.

It is convenient to define the creation and annihilation *super operators* $\hat{a}_j^\dagger$ and $\hat{a}_j$ (notice the hat) acting on $\Gamma_{\underline{v}}$ as follows

$$\hat{a}_j^\dagger \Gamma_{\underline{v}} = \delta_{v_j,0}\pi_j\Gamma_{\underline{v}'}, \tag{A.3}$$

$$\hat{a}_j \Gamma_{\underline{v}} = \delta_{v_j,1}\pi_j\Gamma_{\underline{v}'}, \tag{A.4}$$

where we defined the sign $\pi_j$ as

$$\pi_j := (-1)^{\sum_{r=1}^{j-1} v_j}. \tag{A.5}$$

In (A.3) and (A.4), we defined $v_r' = v_r$ for $r \neq j$ and $v_r' = 1 - v_r$ for $r = j$. The super operators $\hat{a}_j, \hat{a}_j^\dagger$ satisfy the standard anticommutation relations of Dirac fermions. First, it is straightforward to check that

$$a_j\Gamma_{\underline{v}} = (\hat{a}_j^\dagger + \hat{a}_j)\Gamma_{\underline{v}}, \tag{A.6}$$

$$\Gamma_{\underline{v}}a_j = (\hat{a}_j^\dagger - \hat{a}_j)\Gamma_{\underline{v}}. \tag{A.7}$$

In (A.7) we focus on fermionic states with even parity, i.e., for which $\sum_{r=1}^{2L} v_r$ is even. By using (A.6) and (A.7), one can easily derive the commutation relations

$$[a_j, \Gamma_{\underline{v}}] = 2\hat{a}_j\Gamma_{\underline{v}}, \tag{A.8}$$

$$[a_j a_k, \Gamma_{\underline{v}}] = 2(\hat{a}_j^\dagger \hat{a}_k - \hat{a}_k^\dagger a_j)\Gamma_{\underline{v}}. \tag{A.9}$$

For the following, it is convenient to define new fermionic super operators $\hat{a}_{\pm,j}$ as

$$\hat{a}_{\pm,j} := \frac{1}{\sqrt{2}}(\hat{a}_{2m-1} \pm i\hat{a}_{2m}). \tag{A.10}$$

Notice that $\hat{a}_{\pm,j}$ and $\hat{a}_{+,k}$ act as "creation" or "destruction" super operators. For instance, $\hat{a}_{-,k}$ and $\hat{a}_{+,k}$ destroy the operator $c_k^\dagger$ and $c_k$, respectively. Similarly, $\hat{a}_{-,k}^\dagger$ and $\hat{a}_{+,k}^\dagger$ create $c_k^\dagger$ and $c_k$, respectively.

Let us decompose the Liouvillian in (2) in a bulk contribution and in a boundary one as

$$\mathcal{L}(\Gamma_{\underline{\nu}}) = \mathcal{L}_{\text{bulk}}(\Gamma_{\underline{\nu}}) + \mathcal{L}_{\text{boundary}}(\Gamma_{\underline{\nu}}), \tag{A.11}$$

where $\mathcal{L}_{\text{bulk}}$ contains the Hamiltonian part and the dephasing contribution, whereas $\mathcal{L}_{\text{boundary}}$ takes into account the boundary losses. Specifically, the bulk Liouvillian is given as [11,38]

$$\mathcal{L}_{\text{bulk}} = i \sum_{j=1}^{L-1} \sum_{\alpha=\pm} \alpha(\hat{a}_{\alpha,j}^\dagger \hat{a}_{\alpha,j+1} + \hat{a}_{\alpha,j+1}^\dagger \hat{a}_{\alpha,j}) + \frac{\gamma}{2}, \sum_{j=1}^{L} (2\hat{a}_{+,j}^\dagger \hat{a}_{+,j} \hat{a}_{-,j}^\dagger \hat{a}_{-,j} - \hat{a}_{+,j}^\dagger \hat{a}_{+,j} - \hat{a}_{-,j}^\dagger \hat{a}_{-,j}), \tag{A.12}$$

where $\hat{a}_{\pm,j}$ are defined in (A.10). Let us discuss the boundary term $\mathcal{L}_{\text{boundary}}$. Its action on a generic string $\Gamma_{\underline{\nu}}$ reads as

$$\mathcal{L}_{\text{boundary}}(\Gamma_{\underline{\nu}}) = \gamma^- \left( c_1 \Gamma_{\underline{\nu}} c_1^\dagger - \frac{1}{2} \{c_1^\dagger c_1, \Gamma_{\underline{\nu}}\} \right) + c_1 \to c_L, \tag{A.13}$$

where $c_j$ are Dirac fermions. By using (A.6) and (A.7), Eq. (A.13) is rewritten as

$$\mathcal{L}_{\text{boundary}} = -\frac{\gamma^-}{2} \hat{a}_1^\dagger \hat{a}_1 - \frac{\gamma^-}{2} \hat{a}_2^\dagger \hat{a}_2 + i\gamma^- \hat{a}_1^\dagger \hat{a}_2^\dagger + (\hat{a}_1, \hat{a}_2) \to (\hat{a}_{L-1}, \hat{a}_L). \tag{A.14}$$

Finally, by using the definition of $\hat{a}_{+,j}$ and $\hat{a}_{-,j}$ in (A.10), we can rewrite (A.14) as

$$\mathcal{L}_{\text{boundary}} = -\frac{\gamma^-}{2} \hat{a}_{+,1}^\dagger \hat{a}_{+,1} - \frac{\gamma^-}{2} \hat{a}_{-,1}^\dagger \hat{a}_{-,1} - \gamma^- \hat{a}_{-,1}^\dagger \hat{a}_{+,1}^\dagger + \hat{a}_{\pm,1} \to \hat{a}_{\pm,L}. \tag{A.15}$$

Now the first two terms in (A.15) are interpreted as boundary magnetic fields in the Hubbard chain. The last term, however, corresponds to creation of a pair of fermions with opposite spins at the boundary.

Before proceeding, we should observe that to map $\mathcal{L}$ to a Hubbard-like Hamiltonian $H$ as in Ref. [11] we have to perform a unitary transformation as

$$H = i\mathcal{U}^\dagger \mathcal{L} \mathcal{U}, \tag{A.16}$$

where the unitary transformation $\mathcal{U}$ is defined as [11]

$$\mathcal{U} = \prod_{\text{odd } j} (1 - 2\hat{a}_{-,j}^\dagger \hat{a}_{-,j}). \tag{A.17}$$

The effect of (A.17) is to change the sign of the term with $\alpha = -1$ in (A.12). Finally, the Liouvillian is mapped to the Hubbard-like Hamiltonian $H$ as

$$H = -\sum_{j=1}^{L-1} \sum_{\sigma=\uparrow,\downarrow} (c_{j,\sigma}^\dagger c_{j+1,\sigma} + h.c.) + i\gamma \sum_j n_{j,\uparrow} n_{j,\downarrow} - i\frac{\gamma}{2} \sum_j (n_{j,\uparrow} + n_{j,\downarrow})$$
$$- i\frac{\gamma^-}{2} (n_{1,\uparrow} + n_{1,\downarrow} + n_{L,\uparrow} + n_{L,\downarrow}) + i\gamma^- (c_{1,\uparrow}^\dagger c_{1,\downarrow}^\dagger + (-1)^{L+1} c_{L,\uparrow}^\dagger c_{L,\downarrow}^\dagger), \tag{A.18}$$

where we redefined $c_{j,\uparrow} := \hat{a}_{+,j}$ and $c_{j,\downarrow} := \hat{a}_{-,j}$. Now, the Hamiltonian (A.18) is similar to that of the Hubbard chain with boundary magnetic fields. Precisely, Eq. (A.18) describes a chain of spinful fermions with imaginary density-density interaction and imaginary boundary fields. Crucially, the last term in (A.18) describes creation of pairs of fermions with opposite spins at the boundary of the chain. To the best of our knowledge, the boundary pair-production term renders the Hamiltonian (A.18) not integrable.

However, after including a boundary fermion pump term with pump rate $\gamma^+ = \gamma^-$, which is described by the Lindblad operators $L_{x,3} = \sqrt{\gamma^+} c_x^\dagger \delta_{x,1}$ and $L_{x,4} = \sqrt{\gamma^+} c_x^\dagger \delta_{x,L}$, the last term in (A.15) cancels out. As it was observed in Ref. [11], the resulting Hamiltonian is that of the open Hubbard chain with imaginary interactions and imaginary boundary magnetic fields, which is integrable [25].

## A.1   Comparison with the Hubbard model with boundary fields

Although Eq. (A.18) is not integrable in general, as we showed in section 3, the Liouvillian $\mathcal{L}^{(2)}$ can be diagonalized by the Bethe Ansatz. In this section we report the Bethe equations for the open Hubbard chain with boundary magnetic fields. We show that in the two-fermion sector the Bethe equations are the same as the ones derived in section 3 after some appropriate transformations. The fact that the pair creation terms in (A.18) do not affect the Bethe equations could suggest that the model is integrable by using the techniques of Ref. [13].

The Hamiltonian of the one-dimensional Hubbard chain with open boundary conditions reads as [25]

$$
H = -\sum_{j=1}^{L-1}\sum_{\sigma=\uparrow,\downarrow}(c_{j,\sigma}^{\dagger}c_{j+1,\sigma}+h.c.) + 4u\sum_{j}n_{j,\uparrow}n_{j,\downarrow}
$$
$$
-2u\sum_{j}(n_{j,\uparrow}+n_{j,\downarrow}) - p(n_{1,\uparrow}+n_{1,\downarrow}) - p'(n_{L,\uparrow}+n_{L,\downarrow}), \tag{A.19}
$$

where $c_{j,\sigma}$ are spinful fermionic operators, $n_{j,\sigma} := c_{j,\sigma}^{\dagger}c_{j,\sigma}$ is the local fermionic density, $u$ is the interaction strength, and $p, p'$ is the strength of the boundary fields. Eq. (A.19) is the same as (A.18) after redefining $u = i\gamma/4$ and $p = i\gamma^{-}/2$, except for the last term in (A.18). The Bethe equations for the quasimomenta $k_j$ read as [25]

$$
e^{2ik_jL}\frac{e^{ik_j}-p}{1-pe^{ik_j}}\frac{e^{ik_j}-p'}{1-p'e^{ik_j}} = \prod_{\ell=1}^{M}\frac{\sin(k_j)-\lambda_{\ell}+iu}{\sin(k_j)-\lambda_{\ell}-iu}\frac{\sin(k_j)+\lambda_{\ell}+iu}{\sin(k_j)+\lambda_{\ell}-iu}, \tag{A.20}
$$

where $j = 1,\ldots,N$, together with

$$
\prod_{j=1}^{N}\frac{\lambda_{\ell}-\sin(k_j)+iu}{\lambda_{\ell}-\sin(k_j)-iu}\frac{\lambda_{\ell}+\sin(k_j)+iu}{\lambda_{\ell}+\sin(k_j)-iu} = \prod_{m=1,m\neq j}^{M}\frac{\lambda_{\ell}-\lambda_{m}+2iu}{\lambda_{\ell}-\lambda_{m}-2iu}\frac{\lambda_{\ell}+\lambda_{m}+2iu}{\lambda_{\ell}+\lambda_{m}-2iu}, \tag{A.21}
$$

with $\ell = 1,\ldots,M$, and $N, M$ integers. The eigenvalues $\varepsilon$ of the eigenstates are given as

$$
\varepsilon = -\sum_{j=1}^{N}(2\cos(k_j)+2u). \tag{A.22}
$$

Now, we are interested in the case with $N = 2$ and $M = 1$. As it is clear from the case of the tight-binding chain with periodic boundary conditions and bulk dephasing, the spectrum of the Hubbard chain with $N = 2$ and $M = 1$ is mapped to the spectrum of $\mathcal{L}^{(2)}$. To proceed, we can solve (A.22) for $\lambda_1$ to obtain

$$
\lambda_1 = \pm\frac{1}{\sqrt{2}}(\sin^2(k_1)+\sin^2(k_2)+2u^2)^{\frac{1}{2}}. \tag{A.23}
$$

After substituting (A.23) in (A.20), we obtain

$$
e^{2ik_1L}\frac{e^{ik_1}-p}{1-pe^{ik_1}}\frac{e^{ik_1}-p'}{1-p'e^{ik_1}} = \frac{\sin(k_1)-\sin(k_2)+2iu}{\sin(k_1)-\sin(k_2)-2iu}\frac{\sin(k_1)+\sin(k_2)+2iu}{\sin(k_1)+\sin(k_2)-2iu}, \tag{A.24}
$$

$$
e^{2ik_2L}\frac{e^{ik_2}-p}{1-pe^{ik_2}}\frac{e^{ik_2}-p'}{1-p'e^{ik_2}} = \frac{\sin(k_2)-\sin(k_1)+2iu}{\sin(k_2)-\sin(k_1)-2iu}\frac{\sin(k_2)+\sin(k_1)+2iu}{\sin(k_2)+\sin(k_1)-2iu}. \tag{A.25}
$$

After choosing $p = p' = i\gamma^{-}/2$ and $u = i\gamma/4$, Eq. (A.24) and Eq. (A.25) become the same as (18) (19) if one redefines $k_2 \to k_2 + \pi$.

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
