# Peer review of "Free fermions with dephasing and boundary driving: Bethe Ansatz results"

_SciPost Physics Core, doi:SciPost Phys. Core 8, 011 (2025)_

## Round 1 · Referee Report · Anonymous (Referee 1) · 2023-11-17

Strengths

1- The paper presents an interesting new application of the Bethe ansatz to obtain the spectrum in the single-particle sector of a Liouvillian that is not integrable. Based on the results obtained from the Bethe ansatz, the time evolution of the two-point function is studied as an important application.

2- The calculations presented in the paper are technically very challenging and the results are impressive.

3- The presentation is very clear and even though the subject is rather technical, the paper is relatively easy to read.

4- Results obtained from the Bethe ansatz are backed up by excellent agreement with exact numerics.

Weaknesses

1- The progress that is being made in this paper is rather technical than conceptual, and it is unclear to which extent the new results can lead to new physical insights.

Report

The paper presents an interesting new application of the Bethe ansatz to study a fermionic tight-binding chain with bulk dephasing and losses at the boundary, described by a quantum master equation in Lindblad form. While the full Liouvillian is not integrable, there is a closed equation of motion for the equal-time two-point function, and the author employs the Bethe ansatz to determine the spectrum of the superoperator that generates the dynamics of the two-point function. To illustrate the usefulness of the results, diffusive dynamics of a particle that is initially localized at the center of the chain are demonstrated.

The technical developments presented in the paper are impressive. I appreciate very much that even though the subject is very technical, the text is rather accessible. However, I am not convinced that the new results represent a groundbreaking discovery. In particular, it is not clear to me which important novel physical rather than technical insights could arise from the results of the paper. Therefore, I believe that the manuscript would be more suitable for a less selective journal such as SciPost Physics Core.

I do not have substantial criticism regarding the contents of the paper. The points listed below are rather minor.

Requested changes

1- The term "energies" for the eigenvalues of the Liouvillian is somewhat ambiguous. In particular, when "energy levels" are first mentioned in the abstract, it is not clear whether the real or imaginary parts of the energies correspond to decay rates. This should be clarified.

2- In the introduction and below Eq. (26) it is stated that for $\gamma^- = 0$, a specific set of energies is purely imaginary. However, as far as I understand, they are not purely imaginary but have real part $- \gamma$.

3- How should the following statement be interpreted: "For $x_1 = x_2$, $G_{x_1, x_2}$ is given by the first row in (9)." Is it really only the first row or the full term that multiplies the first Heaviside function? Does this statement imply a particular choice of the value $\Theta(0)$?

4- What is the physical meaning of the symmetry $\mathcal{R}$?

5- Maybe I am overlooking something here, but for the ansatz in Eq. (9) it seems to me that $G_{x_2, x_1} = \sigma G_{x_1, x_2}$ without the factor $( - 1)^{x_1 + x_2}$ on the left-hand side.

6- Is inversion symmetry of the setup reflected in the ansatz Eq. (9)?

7- It is not quite clear what is meant with "Let us now impose the “contact” condition obtained by fixing x1 = x2 in (9)." Is this the equality of the values of $G_{x_1, x_2}$ for $x_1$ approaching $x_2$ from above and from below?

8- In the discussion preceding Eq. (26), it is not quite clear where the restrictions on the values of $k_1$ and $k_2$ for finite $\gamma$ come from.

9- In Eq. (26), why is there $L + 1$ in the denominator? To the given order in $L$, the $+ 1$ is negligible.

10- Below Eq. (26), I suppose the reference should be to Fig. 2.

11- The title of Sec. 3.4, "Solutions with vanishing imaginary parts," is somewhat confusing. It would help to clarify that the imaginary parts of the momenta $k_1$ and $k_2$ and not of the eigenvalues $\varepsilon$ vanish.

12- Also in Eqs. (29), (30), and (31), I believe that $L + 1$ can be replaced by $L$.

13- Above Eq. (42), it is stated that "This is obtained by considering the energy $\varepsilon$ with the smallest nonzero real part." Actually, it should be the largest nonzero real part.

14- I find the formulation above Eq. (48), that "it is possible to determine a more convenient choice" for the left eigenvectors, somewhat misleading. For a given eigenvalue, the left eigenvectors are determined by the eigenvalue equation, and there is no choice in how to define them.

15- I believe that in Eq. (63) there is a factor $f(x)$ missing in the sum.

16- Below Eq. (64), there is a typo: "see section (24)."

17- In Sec. 3.5, it does not become quite clear what the string hypothesis actually is or where exactly it is being used. A brief discussion of these points would be helpful.

18- The Liouvillian gap is defined twice, in Eqs. (42) and (77), and the definitions do not agree.

  • validity: top
  • significance: good
  • originality: high
  • clarity: top
  • formatting: perfect
  • grammar: excellent

Author:  Vincenzo Alba  on 2024-11-27  [id 5002]

(in reply to Report 1 on 2023-11-17)

REFEREE:

The progress that is being made in this paper is rather technical than conceptual, and it is unclear to which extent the new results can lead to new physical insights.

However, I am not convinced that the new results represent a groundbreaking discovery. In particular, it is not clear to me which important novel physical rather than technical insights could arise from the results of the paper. Therefore, I believe that the manuscript would be more suitable for a less selective journal such as SciPost Physics Core.

REPLY: I have to disagree with the referee. In the paper I show that it is possible to describe analytically the dynamics of the two-point fermionic correlation function in systems with global dephasing and boundary losses. This is a system of theoretical and experimental interest, as it is clear from the papers investigating the same or similar setups. For instance, very recently Ref. [49,50], which we added to the list of references, explored a similar system mentioning my results, even in the abstract.

REFEREE: 1- The term "energies" for the eigenvalues of the Liouvillian is somewhat ambiguous. In particular, when "energy levels" are first mentioned in the abstract, it is not clear whether the real or imaginary parts of the energies correspond to decay rates. This should be clarified.

REPLY: We thank the referee for this comment. We replaced most of the occurrences of energies with eigenvalues.

REFEREE: In the introduction and below Eq. (26) it is stated that for γ−=0, a specific set of energies is purely imaginary. However, as far as I understand, they are not purely imaginary but have real part −γ.

REPLY: I thank the referee for this remark. Indeed, I have not been very precise. The referee is correct that the real part is -gamma, which originates from the definition of the energy in (11). I changed the manuscript to make this clear.

REFEREE: 3- How should the following statement be interpreted: "For x1=x2, Gx1,x2 is given by the first row in (9)." Is it really only the first row or the full term that multiplies the first Heaviside function? Does this statement imply a particular choice of the value Θ(0)?

REPLY: I thank the referee for this comment. In the manuscript I meant that the case with x_1=x_2 is recovered by keeping only the term multiplying the Theta(x2-x1). Now I changed the text to make this more clear.

REFEREE: What is the physical meaning of the symmetry R?

REPLY: The invariance under the transformation R that I introduced in the manuscript relies on the fact that the Heisenberg equations of motion are linear in G, which in turn relies on the free-fermion nature of the problem. It also relies on hermiticity, which implies a constraint on the amplitudes for the left and right hopping terms in the Hamiltonian. Although it would require a more detailed investigation, it is likely that the symmetry holds for generic linear evolutions of G, which corresponds to a generic hopping, even long range, fermionic Hamiltonian.

REFEREE: Maybe I am overlooking something here, but for the ansatz in Eq. (9) it seems to me that Gx2,x1=σGx1,x2 without the factor (−1)x1+x2 on the left-hand side.

R: We thank the referee for this comment. Indeed, there is a typo in Eq. 6, which I
corrected. Now, it is clear that Eq (6) satisfies (-1)^{x1+x2} G_{x2x1}=sigmaG_{x1x2}.

REFEREE: Is inversion symmetry of the setup reflected in the ansatz Eq. (9)?

REPLY:Inversion symmetry is not directly implemented in the ansatz (6). Indeed, inversion symmetry has to do with the transformation x1->L-x1 and x2->L-x2. Imposing that the ansatz (9) is compatible with inversion symmetry gives constraints on the quasimomenta k1 and k2. So, in summary I didn't use inversion symmetry in the ansatz explicitly. Of course, after substituting the k1 and k2 obtained by solving the Bethe equations, (9) has to be compatible with inversion symmetry. A reason for not imposing inversion symmetry is that the ansatz in (9) should work also in the situation with different loss rates at the edges of the chain, which would break inversion symmetry.

REFEREE: It is not quite clear what is meant with "Let us now impose the “contact” condition obtained by fixing x1 = x2 in (9)." Is this the equality of the values of Gx1,x2 for x1 approaching x2 from above and from below?

REPLY: We thank the referee for this question. The contact condition consists in imposing that the eigenvalue equation for G_{x1x2} holds also in the limit when x1->x2. Indeed, the form of the eigenvalue varepsilon in (11) is obtained by solving (10) for x1ne x2. However, one has to require that (10) holds with the same varepsilon (11) also for x1=x2. We now explain that better in the manuscript.

REFEREE: In the discussion preceding Eq. (26), it is not quite clear where the restrictions on the values of k1 and k2 for finite γ come from.

REPLY: The restrictions on the values of k1 and k2 comes from the properties of the Bethe equations discusse after (18,19). For instance, one has to discard the pairs k1 k2 such that k1+k2=0mod pi because they would lead to a vanishing eigenvector. We modified the manuscript to stress that the conditions are the ones discussed at the beginning of Section 3.2.

REFEREE: In Eq. (26), why is there L+1 in the denominator? To the given order in L, the +1 is negligible.

REPLY: Although the referee is correct we prefer to keep the L+1 in (26). The reason is that otherwise there will be a 1/L correction on the real part in Eq. (27-28), whereas now the 1/L correction is only on the imaginary part.

REFEREE: Below Eq. (26), I suppose the reference should be to Fig. 2.

REPLY:Yes, the referee is correct. We fixed the typo.

REFEREE: The title of Sec. 3.4, "Solutions with vanishing imaginary parts," is somewhat confusing. It would help to clarify that the imaginary parts of the momenta k1 and k2 and not of the eigenvalues ε vanish.

REPLY: We modified the title of the section to stress that it is the imaginary parts of k1 of k2 that vanish.

REFEREE: Also in Eqs. (29), (30), and (31), I believe that L+1 can be replaced by L.

REPLY: As for the previous point of the referee, we prefer to keep the L+1 in the denominator.

REFEREE: Above Eq. (42), it is stated that "This is obtained by considering the energy ε
with the smallest nonzero real part." Actually, it should be the largest nonzero real part.

REPLY: We thank the referee for spotting this typo. We corrected it.

REFEREE: I find the formulation above Eq. (48), that "it is possible to determine a more convenient choice" for the left eigenvectors, somewhat misleading. For a given eigenvalue, the left eigenvectors are determined by the eigenvalue equation, and there is no choice in how to define them.

REPLY: Following the referee suggestions we simplified the discussion about the left and right eigenvectors.

REFEREE: I believe that in Eq. (63) there is a factor f(x)
missing in the sum.

REPLY: We corrected the typo.

REFEREE: Below Eq. (64), there is a typo: "see section (24)."

REPLY: We corrected the typo.

REFEREE:
In Sec. 3.5, it does not become quite clear what the string hypothesis actually is or where exactly it is being used. A brief discussion of these points would be helpful.

REPLY: As it is discussed in section 3.3 roughly all the solutions k1,k2 of the Bethe equations have vanishing imaginary parts in the limit Ltoinfty. However, there are L sets of solutions that have finite imaginary parts for Ltoinfty. Since k1,k2 have to be complex conjugated, when plotted in the plane Im k versus Re k, as in Fig. 2, they form a string'' in the plane. So far this is exact. However, in the limit Ltoinfty one can simplify the Bethe equations. As discussed at page 11, one obtains Eqs (35-36). These equations are approximate, and hence their solution provides an approximation for k1,k2. However, the deviations from the exact solutions of the Bethe equations vanish exponentially in the limit Ltoinfty. This is what we refer to as the string hypothesis because the reasoning employed is similar to the standard string hypothesis in spin chain. We added another sentence below Eq(34) to stress that.

REFEREE: The Liouvillian gap is defined twice, in Eqs. (42) and (77), and the definitions do not agree

REPLY: I thank the referee for spotting the typo. I corrected it.

---

## Round 1 · Referee Report · Anonymous (Referee 2) · 2023-12-2

Strengths

1 - very comprehensive generalization of recent results on Bethe ansatz results for open quantum systems 2- new method for explicitly computing 2 point correlators

Report

The author studies a free-fermion chain subjected to bulk Markovian local dephasing and boundary loss. The full Liouvillian is seemingly not integrable. Despite this, the author manages to use Bethe ansatz to solve the 2-point correlation dynamics, which I find very interesting. I find the work both conceptually and technically innovative and recommend publication, but I would ask the author to address the below question.

Requested changes

1 - Is the solvability of the pure loss boundaries related to the triangular form of the Liouvillian as exploited in [13,17]? It would be nice for the author to comment on this. 2- If so, would this approach work for bulk loss?

  • validity: top
  • significance: good
  • originality: high
  • clarity: high
  • formatting: excellent
  • grammar: excellent

Author:  Vincenzo Alba  on 2024-11-27  [id 5001]

(in reply to Report 2 on 2023-12-02)

REFEREE: The author studies a free-fermion chain subjected to bulk Markovian local dephasing and boundary loss. The full Liouvillian is seemingly not integrable. Despite this, the author manages to use Bethe ansatz to solve the 2-point correlation dynamics, which I find very interesting. I find the work both conceptually and technically innovative and recommend publication, but I would ask the author to address the below question.

REPLY: I thank the referee for the positive evaluation.

REFEREE: Is the solvability of the pure loss boundaries related to the triangular form of the Liouvillian as exploited in [13,17]? It would be nice for the author to comment on this.

REPLY: I thank the referee for this comment. Indeed, I think that the solvability of the model could be understood using the results of Ref. [13,17]. One hint that this is the case is that, as I show in the Appendix, the Bethe equations that I derive are the same as those for the Hubbard model with imaginary interaction and imaginary boundary fields, which is integrable. However, the Liouvillian contains an extra pair creation term with imaginary strength, which does not alter the Bethe equations.

REFEREE: If so, would this approach work for bulk loss?

REPLY: As for the case with bulk loss and bulk dephasing, I think that it will be solvable with the method although I think that the result will be trivial, meaning that the spectrum of the Liouvillian will be the same as without loss except for a shift of the eigenvalues.

---

## Round 2 · Author Response

Dear Editors,

In the following I reply to the comments of the referees and highlight the main changes in the manuscript.

Sincerely,

Vincenzo Alba

REFEREE I

REFEREE: The author studies a free-fermion chain subjected to bulk Markovian local dephasing and boundary loss. The full Liouvillian is seemingly not integrable. Despite this, the author manages to use Bethe ansatz to solve the 2-point correlation dynamics, which I find very interesting. I find the work both conceptually and technically innovative and recommend publication, but I would ask the author to address the below question.

REPLY: I thank the referee for the positive evaluation.

REFEREE: Is the solvability of the pure loss boundaries related to the triangular form of the Liouvillian as exploited in [13,17]? It would be nice for the author to comment on this.

REPLY: I thank the referee for this comment. Indeed, I think that the solvability of the model could be understood using the results of Ref. [13,17]. One hint that this is the case is that, as I show in the Appendix, the Bethe equations that I derive are the same as those for the Hubbard model with imaginary interaction and imaginary boundary fields, which is integrable. However, the Liouvillian contains an extra pair creation term with imaginary strength, which does not alter the Bethe equations.

REFEREE: If so, would this approach work for bulk loss?

REPLY: As for the case with bulk loss and bulk dephasing, I think that it will be solvable with the method although I think that the result will be trivial, meaning that the spectrum of the Liouvillian will be the same as without loss except for a shift of the eigenvalues.

REFEREE II

REFEREE:

The progress that is being made in this paper is rather technical than conceptual, and it is unclear to which extent the new results can lead to new physical insights.

However, I am not convinced that the new results represent a groundbreaking discovery. In particular, it is not clear to me which important novel physical rather than technical insights could arise from the results of the paper. Therefore, I believe that the manuscript would be more suitable for a less selective journal such as SciPost Physics Core.

REPLY: I have to disagree with the referee. In the paper I show that it is possible to describe analytically the dynamics of the two-point fermionic correlation function in systems with global dephasing and boundary losses. This is a system of theoretical and experimental interest, as it is clear from the papers investigating the same or similar setups. For instance, very recently Ref. [49,50], which we added to the list of references, explored a similar system mentioning my results, even in the abstract.

REFEREE: 1- The term "energies" for the eigenvalues of the Liouvillian is somewhat ambiguous. In particular, when "energy levels" are first mentioned in the abstract, it is not clear whether the real or imaginary parts of the energies correspond to decay rates. This should be clarified.

REPLY: We thank the referee for this comment. We replaced most of the occurrences of energies with eigenvalues.

REFEREE: In the introduction and below Eq. (26) it is stated that for γ−=0, a specific set of energies is purely imaginary. However, as far as I understand, they are not purely imaginary but have real part −γ.

REPLY: I thank the referee for this remark. Indeed, I have not been very precise. The referee is correct that the real part is -gamma, which originates from the definition of the energy in (11). I changed the manuscript to make this clear.

REFEREE: 3- How should the following statement be interpreted: "For x1=x2, Gx1,x2 is given by the first row in (9)." Is it really only the first row or the full term that multiplies the first Heaviside function? Does this statement imply a particular choice of the value Θ(0)?

REPLY: I thank the referee for this comment. In the manuscript I meant that the case with x_1=x_2 is recovered by keeping only the term multiplying the Theta(x2-x1). Now I changed the text to make this more clear.

REFEREE: What is the physical meaning of the symmetry R?

REPLY: The invariance under the transformation R that I introduced in the manuscript relies on the fact that the Heisenberg equations of motion are linear in G, which in turn relies on the free-fermion nature of the problem. It also relies on hermiticity, which implies a constraint on the amplitudes for the left and right hopping terms in the Hamiltonian. Although it would require a more detailed investigation, it is likely that the symmetry holds for generic linear evolutions of G, which corresponds to a generic hopping, even long range, fermionic Hamiltonian.

REFEREE: Maybe I am overlooking something here, but for the ansatz in Eq. (9) it seems to me that Gx2,x1=σGx1,x2 without the factor (−1)x1+x2 on the left-hand side.

R: We thank the referee for this comment. Indeed, there is a typo in Eq. 6, which I

corrected. Now, it is clear that Eq (6) satisfies (-1)^{x1+x2} G_{x2x1}=sigmaG_{x1x2}.

REFEREE: Is inversion symmetry of the setup reflected in the ansatz Eq. (9)?

REPLY:Inversion symmetry is not directly implemented in the ansatz (6). Indeed, inversion symmetry has to do with the transformation x1->L-x1 and x2->L-x2. Imposing that the ansatz (9) is compatible with inversion symmetry gives constraints on the quasimomenta k1 and k2. So, in summary I didn't use inversion symmetry in the ansatz explicitly. Of course, after substituting the k1 and k2 obtained by solving the Bethe equations, (9) has to be compatible with inversion symmetry. A reason for not imposing inversion symmetry is that the ansatz in (9) should work also in the situation with different loss rates at the edges of the chain, which would break inversion symmetry.

REFEREE: It is not quite clear what is meant with "Let us now impose the “contact” condition obtained by fixing x1 = x2 in (9)." Is this the equality of the values of Gx1,x2 for x1 approaching x2 from above and from below?

REPLY: We thank the referee for this question. The contact condition consists in imposing that the eigenvalue equation for G_{x1x2} holds also in the limit when x1->x2. Indeed, the form of the eigenvalue varepsilon in (11) is obtained by solving (10) for x1ne x2. However, one has to require that (10) holds with the same varepsilon (11) also for x1=x2. We now explain that better in the manuscript.

REFEREE: In the discussion preceding Eq. (26), it is not quite clear where the restrictions on the values of k1 and k2 for finite γ come from.

REPLY: The restrictions on the values of k1 and k2 comes from the properties of the Bethe equations discusse after (18,19). For instance, one has to discard the pairs k1 k2 such that k1+k2=0mod pi because they would lead to a vanishing eigenvector. We modified the manuscript to stress that the conditions are the ones discussed at the beginning of Section 3.2.

REFEREE: In Eq. (26), why is there L+1 in the denominator? To the given order in L, the +1 is negligible.

REPLY: Although the referee is correct we prefer to keep the L+1 in (26). The reason is that otherwise there will be a 1/L correction on the real part in Eq. (27-28), whereas now the 1/L correction is only on the imaginary part.

REFEREE: Below Eq. (26), I suppose the reference should be to Fig. 2.

REPLY:Yes, the referee is correct. We fixed the typo.

REFEREE: The title of Sec. 3.4, "Solutions with vanishing imaginary parts," is somewhat confusing. It would help to clarify that the imaginary parts of the momenta k1 and k2 and not of the eigenvalues ε vanish.

REPLY: We modified the title of the section to stress that it is the imaginary parts of k1 of k2 that vanish.

REFEREE: Also in Eqs. (29), (30), and (31), I believe that L+1 can be replaced by L.

REPLY: As for the previous point of the referee, we prefer to keep the L+1 in the denominator.

REFEREE: Above Eq. (42), it is stated that "This is obtained by considering the energy ε

with the smallest nonzero real part." Actually, it should be the largest nonzero real part.

REPLY: We thank the referee for spotting this typo. We corrected it.

REFEREE: I find the formulation above Eq. (48), that "it is possible to determine a more convenient choice" for the left eigenvectors, somewhat misleading. For a given eigenvalue, the left eigenvectors are determined by the eigenvalue equation, and there is no choice in how to define them.

REPLY: Following the referee suggestions we simplified the discussion about the left and right eigenvectors.

REFEREE: I believe that in Eq. (63) there is a factor f(x)

missing in the sum.

REPLY: We corrected the typo.

REFEREE: Below Eq. (64), there is a typo: "see section (24)."

REPLY: We corrected the typo.

REFEREE:

In Sec. 3.5, it does not become quite clear what the string hypothesis actually is or where exactly it is being used. A brief discussion of these points would be helpful.

REPLY: As it is discussed in section 3.3 roughly all the solutions k1,k2 of the Bethe equations have vanishing imaginary parts in the limit Ltoinfty. However, there are L sets of solutions that have finite imaginary parts for Ltoinfty. Since k1,k2 have to be complex conjugated, when plotted in the plane Im k versus Re k, as in Fig. 2, they form a ``string'' in the plane. So far this is exact. However, in the limit Ltoinfty one can simplify the Bethe equations. As discussed at page 11, one obtains Eqs (35-36). These equations are approximate, and hence their solution provides an approximation for k1,k2. However, the deviations from the exact solutions of the Bethe equations vanish exponentially in the limit Ltoinfty. This is what we refer to as the string hypothesis because the reasoning employed is similar to the standard string hypothesis in spin chain. We added another sentence below Eq(34) to stress that.

REFEREE: The Liouvillian gap is defined twice, in Eqs. (42) and (77), and the definitions do not agree

REPLY: I thank the referee for spotting the typo. I corrected it.

---

## Editorial Decision

published